## RESEARCH ARTICLE

# Invasive goldfish (*Carassius auratus*) maintain aerobic scope across acute warm water temperatures

Nazeefa A. Nashrah[1,*], Nicholas E. Mandrak[1,2] and Melanie D. Massey[1]

## ABSTRACT

Goldfish (*Carassius auratus*) were first introduced to the Laurentian Great Lakes when it was first introduced into Lake Ontario in the 1800s. In the past 15 years, there have been dramatic increases in both goldfish abundance and geographic spread across North America, including the Great Lakes, raising concerns about its potential for negative impacts on aquatic ecosystems. Climate studies suggest that habitat ranges suitable for goldfish will continue to expand in the future as water temperatures rise towards its thermal optima. We explore whether warmer temperatures are physiologically suitable for a population of wild, invasive goldfish (Hamilton, ON, Canada) by testing aerobic scope (AS) at current (26°C) and predicted (30°C) peak summer water temperatures. Goldfish were first acclimated to a common-garden average summer pond temperature (22°C), then their AS was estimated by calculating the difference between routine and maximum metabolic rates (RMR and MMR) at the two test temperatures. Our results demonstrate that wild goldfish sustain their AS through increases in both RMR and MMR from 26°C to 30°C (mass-standardized means of 1.07 versus 1.17 $mgO_2h^{-1}$ at 26°C versus 30°C, respectively). This ability to sustain aerobic energy budget at high peak water temperatures could offer physiological benefits to this invasive population in a warming climate.

KEY WORDS: Climate change, Metabolic rate, Respirometry, Thermal performance

## INTRODUCTION

A major cause of freshwater biodiversity loss is the introduction of invasive species (Sala et al., 2000; Simberloff, 2020). The global rate of invasive freshwater fish introductions has increased rapidly since 1980 (Gozlan et al., 2010) and, since the European colonization, the introduction of non-native species has caused broad-scale homogenization of fish faunas across North America, which has had substantial ecological and evolutionary consequences (Anas and Mandrak, 2021). Non-native fishes in North America include species imported from other continents and species native to North America but introduced beyond their historical zoogeographic boundaries

[1]Department of Biological Sciences, University of Toronto, Scarborough, ON M1C 1A4, Canada. [2]Department of Physical and Environmental Sciences, University of Toronto, Scarborough, ON M1C 1A4, Canada.

*Author for correspondence (nazeefa.nashrah@mail.utoronto.ca)

N.A.N., 0009-0004-6441-3231; N.E.M., 0000-0001-8335-9681; M.D.M., 0000-0002-9036-315X

through authorized and unauthorized pathways (Anas and Mandrak, 2021). Such biological invasions occur in sequential phases that consist of propagule introduction into a new ecosystem, establishment of a self-sustaining population, and spread into uncolonized ecosystems (Liu et al., 2015).

Throughout the past century, the Laurentian Great Lakes have been severely impacted both ecologically and economically by the establishment of self-sustaining populations of aquatic invasive species (Mandrak and Cudmore, 2010; Lodge et al., 2016) and, as anthropogenic climate change increases water temperatures, there is an increased potential for further invasion by non-native, warmwater species (Stachowicz et al., 2002; Hubbard et al., 2023). In addition to enhancing invasion pathways, climate change can further increase the invasiveness of non-native species as habitat suitability increases to match physiological optima (Hubbard et al., 2024; Johnson et al., 2009; Quattrocchi et al., 2023; Rahel and Olden, 2008; Stachowicz et al., 2002). Invasive freshwater fishes have also been found to outcompete native species in climate-change contexts, with warmer temperatures minimally affecting or even benefiting competitive performance relative to native comparators (Kłosiński et al., 2024; Kuczynski et al., 2018). To inform invasive species management, there is a great need to understand how the performance of invaders will change as the climate changes (Beaury et al., 2020).

In relation to the ambient environment and complex interactions among ecological factors (e.g. predators, competitors, parasites), establishment in one ecosystem and subsequent secondary spread is often limited by the organism's physiology (Behrens et al., 2017). One factor that may mechanistically influence establishment and spread of invasive species is the aerobic scope (AS), which has been associated with numerous fitness-related parameters (e.g. energy available for reproduction, digestion, growth, movement) (Claireaux and Lefrançois, 2007; Chabot et al., 2016; Behrens et al., 2017). AS represents an animal's temperature-dependent capacity to increase its aerobic metabolic rate above basal levels (Fry, 1947; Halsey et al., 2018). Although AS is mainly calculated from the difference between the standard metabolic rate (SMR), representing the minimum level of metabolism needed for basic homeostasis, and maximum metabolic rate (MMR), representing the highest level of aerobic metabolism under given environmental conditions (Svendsen et al., 2016; Roche et al., 2013), some studies have also estimated AS as the difference between the routine metabolic rate (RMR), representing the rate of oxygen consumption when a non-quiescent animal is undertaking minor movements, and the MMR (Killen et al., 2014; Poletto et al., 2017; Chabot et al., 2016; Metcalfe et al., 2016).

AS and ecological thermal performance have been theoretically tied in the oxygen- and capacity-limited thermal tolerance (OCLTT) hypothesis (Pörtner, 2001; Pörtner et al., 2017). The hypothesis suggests that as temperatures approach biologically limiting values, an animal's ability to supply oxygen to tissues is restricted, reducing performance (Pörtner, 2001). As such, the hypothesis suggests that

animals optimize fitness-related performance when living within temperature ranges where AS is maximized with diminishing performance as AS decreases at higher/lower temperatures (Pörtner, 2001). The hypothesis thus implies that critical performances are linked causally with AS and animals should have optimal fitness at a particular temperature (Pörtner, 2001). Although the OCLTT hypothesis is argued to be applicable broadly across animal taxa (Verberk et al., 2016), recent studies examining acute responses to temperature changes have questioned the generality of the OCLTT hypothesis for aquatic animals (Clark et al., 2013; Ern et al., 2014; Norin et al., 2014; Wang et al., 2014) prompting a debate on whether the OCLTT hypothesis can be used as an unifying model to understand thermal tolerance of animals (Verberk et al., 2016).

Studies have proposed AS as a useful tool to characterize fish habitat suitability (Teal et al., 2018) and predict migratory success (Farrell et al., 2008). Given that thermal metabolic performance metrics (AS, RMR, MMR) have been used to explain species distributions, they may also be useful in the context of invasion biology (Christensen et al., 2021), including evaluating invader performance and resilience to global warming (Marras et al., 2015). Although uncommonly used in invasion biology studies, AS has been used to model habitat suitability (Quattrocchi et al., 2023) and to infer the competitive performance (Christensen et al., 2021) of the invasive round goby (*Neogobius melanostomus*) under climate-change conditions. Aerobic performance is also associated with a number of characteristics that are strongly predictive of invasion success and impacts, including food uptake, locomotor ability, boldness, competitive dominance, and territorial aggression (Killen et al., 2014; Auer et al., 2015b; Dick et al., 2017). Understanding how invader metabolism may be altered under climate-change conditions may therefore provide insights about future performance.

Goldfish (*Carassius auratus*) is a globally invasive freshwater fish species whose invasion appears to be rapidly escalating under climate change (Massey et al., 2025). Native to eastern Asia and closely related to the common carp (*Cyprinus carpio*), goldfish has been introduced to the Laurentian Great Lakes intentionally through stocking and release from the ornamental fish trade and the commercial baitfish industry (Boston et al., 2023; Massey et al., 2025). As an invader, goldfish has been shown to alter aquatic ecosystems through increased turbidity and nutrient mobilization, which negatively impacts native fishes (Matsuzaki et al., 2009) through competition, predation, habitat degradation and lower water quality (Van Zuiden and Sharma, 2016; Massey et al., 2025). Several studies have suggested that by regulating growth and reproductive activities to adapt to new environments, goldfish is able to increase its rate of successful colonization (Tarkan et al., 2010; Liu et al., 2015). Such invasive characteristics that facilitate colonization of new environments tend to arise from physiologically regulated life-history traits (e.g. changes in body size and fecundity) (Jia, 2019) and can provide an advantage for individual fitness when environmental conditions change (Bolnick et al., 2011; Auer et al., 2015a).

Goldfish is also a long-standing physiological model in thermal biology but is rarely studied in ecophysiology or invasion biology contexts (Boston et al., 2023; Massey et al., 2025). Many studies to date have investigated the thermal metabolism of commercially reared (pet or aquaculture) strains. For example, Fry and Hart (1948a) conducted foundational experiments evaluating the thermal AS of commercial goldfish, demonstrating peak performance in AS around 28°C, with reductions in performance as temperatures moved away from this optimum. Ferreira et al. (2014) found that goldfish acclimated to warm temperatures (20°C and 30°C) had

optimal aerobic performance near their respective acclimation temperatures with reductions in performance above and below these temperatures. However, when cool (12°C) acclimation temperatures were used, AS was maintained and optimal across a plateau of temperatures from 11–30°C (Ferreira et al., 2014). Together, these studies suggest that AS in goldfish is generally optimal at high temperatures, but can be flexibly influenced by prior conditions, at least in these commercially acquired fish. However, it is currently unknown how invasive goldfish, which may differ due to adaptation and acclimation to natural conditions (e.g. with seasonal variation in temperature) may vary in their AS.

In the present study, we aimed to evaluate the AS of invasive, wild-caught goldfish, sourced from an urban pond in Hamilton, Ontario, Canada, when tested acutely at current (26°C) and +4°C scenario (30°C) peak summer pond temperatures. Our goal was to understand how their performance, measured via AS, changes under ecologically relevant short-term warming events. Urban-pond ecosystems in Canada frequently harbor abundant invasive goldfish (Chan et al., 2019; Massey et al., 2025) and are characterized by warm, cycling thermal profiles (Chiandet and Xenopoulos, 2016), with average summer mean temperatures of 22°C and maximum temperatures of 26°C at our study site (Fisheries and Oceans Canada, pers. comm.). Urban-pond goldfish represent high-risk populations for dispersal into nearby waterbodies, such as Lake Ontario, due to intense human-release pressure (Chan et al., 2019), but almost nothing is known about their ecophysiology (Massey et al., 2025). This study represents a first glance at the ecophysiological responses of urban, invasive goldfish to climate change conditions.

## RESULTS

The average RMR, standardized to an average 5.76 g fish, was 1.55 mgO$_2$h$^{-1}$ at 26°C and 1.80 mgO$_2$h$^{-1}$ at 30°C (Fig. 1). The average MMR, standardized to a 5.76 g fish, was 2.06 mgO$_2$h$^{-1}$ at 26°C and 2.35 mgO$_2$h$^{-1}$ at 30°C (Fig. 1). The average AS, standardized to an average 5.76 g fish, was 1.07 mgO$_2$h$^{-1}$ at 26°C and 1.17 mgO$_2$h$^{-1}$ at 30°C (Fig. 1).

Because RMR, MMR, and AS values were log-transformed to aid in interpreting model coefficients, effect size estimates for test temperature have been back-transformed (exponentiated) and can be interpreted multiplicatively (i.e. a factorial percent increase or decrease). Model estimates for the mass covariate have not been back-transformed as mass was log-transformed and thus already scales proportionally with RMR, MMR, and AS (i.e. as a proportionate percent change). Results are presented with coefficient estimates and 90% uncertainty intervals (UIs).

Based on the Bayesian models controlling for mass, there were increases in RMR and MMR, but not AS, when fish were tested at 30°C versus 26°C (Fig. 2). The average RMR at the 30°C test temperature was 12.9% higher than at the 26°C test temperature [Fig. 2; median: 1.13, 90% UIs: (0.79, 1.52)]. The average MMR at the 30°C test temperature was 10.0% higher than at the 26°C test temperature [Fig. 2; median: 1.10, 90% UIs: (0.89, 1.41)]. The average AS was 0.0% higher at the 30°C test temperature than at the 26°C test temperature [Fig. 2; median: 1.00, 90% UIs: (0.64, 1.58)].

The mass covariate estimates differed between RMR, MMR, and AS trials, but mass scaled positively with oxygen consumption in all cases (Fig. 2). For RMR, on average, a 1% increase in mass led to a 0.41% increase in RMR [median: 0.41, 90% UIs: (0.08, 0.86)]. For MMR, on average, a 1% increase in mass led to a 0.57% increase in MMR [median: 0.57, 90% UIs: (0.24, 0.93)]. For AS, on average, a

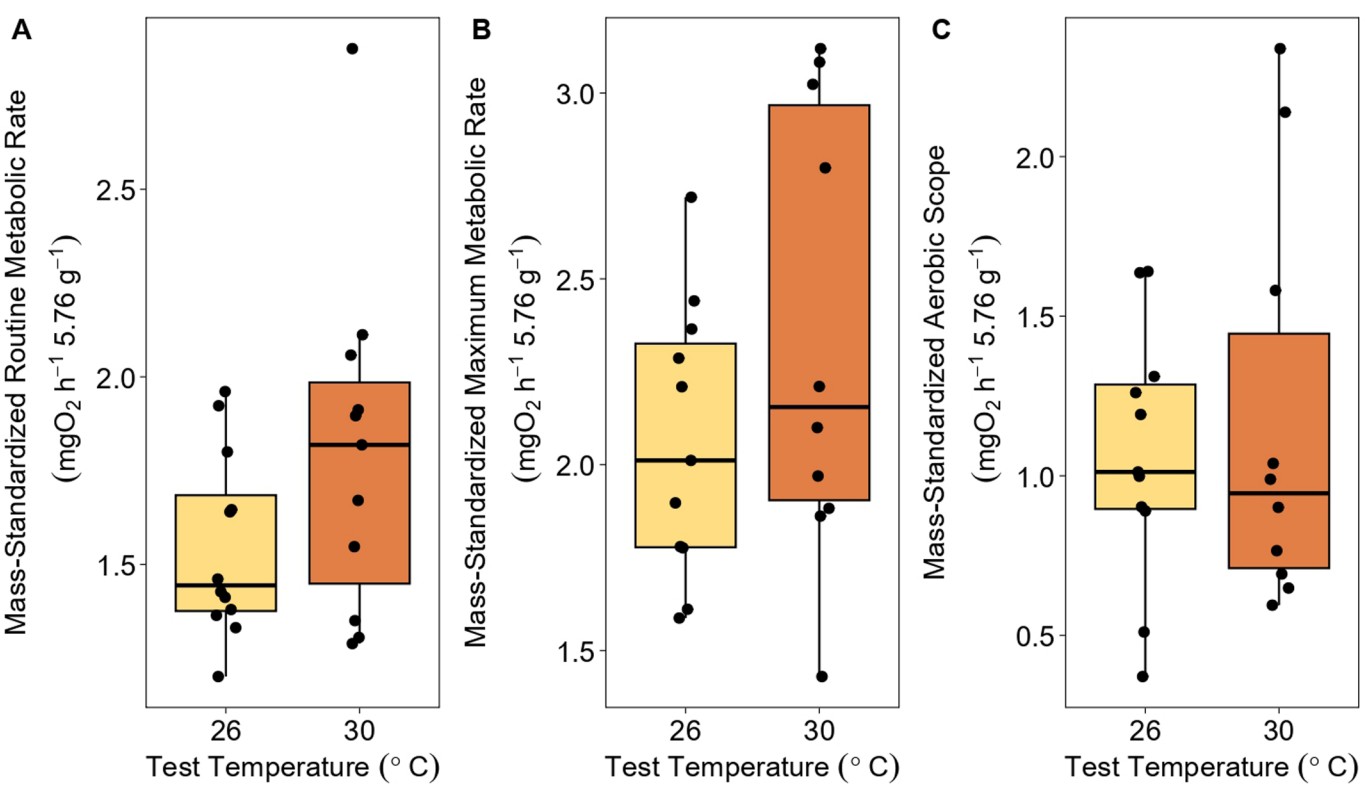

**Fig. 1. Comparison of RMR, MMR, and AS tested acutely at 26°C or 30°C.** Boxplots depicting the median mass-adjusted (A) RMR, (B) MMR, and (C) AS of wild-caught invasive goldfish (*C. auratus*) tested at 26°C (*N*=11) and 30°C (*N*=10), standardized to the mass of an average fish in the experiment (5.76 g). Points represent individual fish.

1% increase in mass led to a 0.76% increase in AS [median: 0.76, 90% UIs: (0.07. 1.46)].

## DISCUSSION

Average global water temperatures are increasing under climate change and recent projections predict increased occurrences of

extreme heat waves (Meehl and Tebaldi, 2004; Perkins et al., 2012; Seneviratne et al., 2014). Despite rising concerns regarding the spread of goldfish in the wild (Massey et al., 2025), and a large body of work investigating the physiology of commercial goldfish (e.g. Ferreira et al., 2014; Fry and Hart, 1948a,b; Herrera-Castillo et al., 2024), little is known about how climate-change conditions, such as

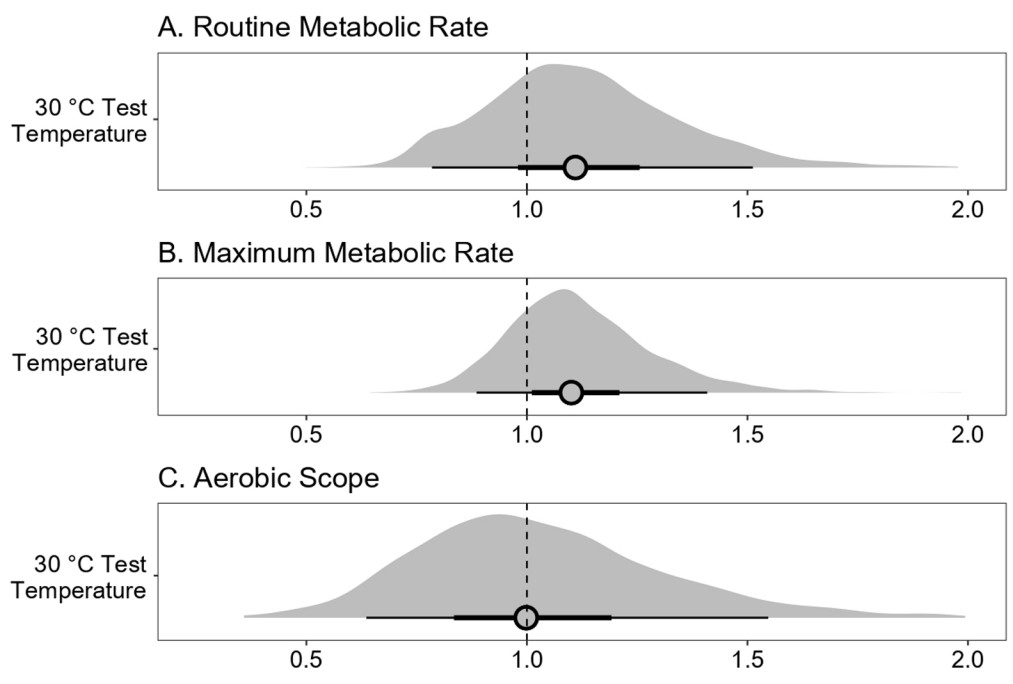

**Fig. 2. Model effect sizes representing changes in RMR, MMR, and AS tested acutely at 30°C, relative to 26°C.** Posterior effect size estimates of a model examining the effects of an acute 30°C test temperature on the (A) RMR, (B) MMR, and (C) AS of wild-caught invasive goldfish (*C. auratus*) relative to a 26°C test temperature. Posterior effect sizes are represented multiplicatively (e.g. a value of 1.10 is equivalent to a 10% increase in metabolism at 30°C, relative to 26°C). Distributions represent the full variation in posterior effect sizes; point interval estimates are beneath, with posterior medians (filled circles), thick lines (50% uncertainty intervals), and thin lines (90% uncertainty intervals).

heat anomalies, impact the ecophysiology of wild goldfish. Here, we show that invasive urban-pond goldfish are able to maintain stable AS under acute current and predicted maximum pond temperatures of 26°C and 30°C, suggesting their metabolic performance is not compromised under increasingly hot acute thermal exposures common in their introduced environments.

Species that can acclimate rapidly and remodel their physiology to compensate for the effects of temperature variation are expected to acquire increasingly greater energetic and competitive advantages during a more thermally variable future (Sandblom et al., 2014; Seebacher et al., 2014). This flexibility may also concomitantly advantage the establishment and spread of invasive species (Davidson et al., 2011); indeed, maintenance of AS at high temperatures has been described in the highly invasive round goby (Behrens et al., 2017; Christensen et al., 2021). Given that there is evidence to suggest invasive species may have higher mass-specific metabolic rates than native species (Lagos et al., 2017) and AS explains individual variation in feeding rate, a key driver of invader impacts and spread (Auer et al., 2015b; Dick et al., 2017), it follows that maintaining high AS under hot temperatures may provide benefits to invasive goldfish under climate-change conditions. At the very least, our results suggest invasive goldfish have the ability to maintain a high capacity for foraging, growth, and reproduction in challenging urban pond environments under current and future scenarios, given the extreme thermal cycling and stratification that occurs in these ponds (Chiandet and Xenopoulos, 2016; Marques and Mandrak, 2024).

The maintenance of AS that we observed across warm temperatures could be due, at least partly, to seasonal plasticity as a result of the previous thermal experience of our study goldfish. Our results align with those of Ferreira et al. (2014), who found that acclimation to cool temperatures in commercial goldfish led to a maintenance of AS across a broad range of temperatures (11–30°C). Although our study fish were acclimated to average summer temperatures for a 3-week period in the lab, they previously experienced ambient, seasonal winter temperatures (6°C average), which may have led to this similar pattern of AS maintenance across 26–30°C. The possibility of cold acclimation leading to broad maintenance of optimal AS was explored in a closely related species, crucian carp (*Carassius carassius*), which develops a broad optimal temperature for AS window while entering, and recovering from, winter hibernation (Vornanen et al., 2009; 2011). This response is thought to provide the crucian carp with advantages towards exploiting the broadest possible thermal niche for foraging (Vornanen et al., 2009, 2011). To fully explore this idea, future studies should employ a broader range of acute test temperatures to examine the full temperature-AS reaction norm of wild goldfish at different times of the year.

There are several limitations to our study. First, while swim-flume and chase methods are widely accepted, the chase method that we used can yield inconsistent results due to its intrinsic variability and potential to induce handling stress (Raby et al., 2020). However, the chase method as a proxy of true MMR remains suitable as its measurement error likely lacks thermal bias (Raby et al., 2020). Therefore, it is worth noting that, although our results may underestimate the AS values, this under-estimation is expected to be consistent across both temperatures. Second, we used RMR, rather than SMR, in our estimation of AS. The use of RMR, which includes the minor costs of activity (i.e. in a fish that is not fully quiescent), may ultimately have led to underestimations of AS. Third, as a modified closed-respirometry technique was employed in our study, there is a possibility that fish could have been disturbed

when valves were closed to cease flushing, increasing our estimate of RMR, and again, leading to an underestimation of AS. Therefore, future studies could employ the preferred intermittent flow respirometry method that combines features of closed and flow-through respirometry while reducing problems associated with both techniques (Svendsen et al., 2016). However, taken together, these limitations suggest our estimates of AS are conservative, rather than exaggerated.

Additionally, we tested all individuals first at 26°C, then at 30°C. Because the order of test temperatures was not randomized, it is possible that fish acclimated to the earlier test temperature affected later measurements at 30°C. Although this may be somewhat ecologically realistic, assuming in the wild that acute heat exposures are more likely to increase in temperature sequentially. We also rapidly acclimated fish from common-garden temperatures to test temperatures during trials, which may have induced acute thermal stress. Therefore, our results are reflective of how fish metabolism may respond to rapid changes in temperature (e.g. a heat shock), rather than more gradual warming over the course of hours or days. Future study designs should consider randomizing treatments when individuals are repeatedly measured and gradual exposure to acute test temperature at a predetermined rate.

It is also worth noting that variation in physiological thermal performance of our study fish could have been attributable to factors not studied in the present experiment. Local genetic adaptation, developmental plasticity, and even transgenerational plasticity to pond conditions may have influenced the responses of our study fish (West-Eberhard, 2003; Massey and Hutchings, 2021). These alternative sources of variation in AS highlight the need to examine different populations of goldfish with differing ecological experiences. For example, in the context of relative risk assessment for invasive species, comparator populations may include wild fish from larger waterbodies including lakes and other ponds with differing abiotic characteristics.

As goldfish abundance and spread increases in North America, it is critical that we understand its ecophysiology to predict its spread and impacts (Massey et al., 2025). Because most fundamental constraints on species distribution occur due to physiological limitations and performance in response to the ambient environment, changes in physiological response in relation to the ambient environment must be accounted for to predict the distribution and potential impacts posed by fish invasions (Pörtner and Peck, 2010). Our results suggest that, even as maximum daily water temperatures increase, wild goldfish will be able to sustain the energy demands of important biological processes including growth and reproduction specifically at remarkably high acute water temperatures of 26°C and 30°C. Ultimately, this flexibility in thermal capacity may benefit invasive goldfish as climate change progresses (Ford and Beitinger, 2005; Sandblom et al., 2014).

## MATERIALS AND METHODS

All holding and experimental protocols were approved by the University of Toronto Biological Sciences Local Animal Care Committees (protocol no. 20012759).

### Fish collection and husbandry

A flowchart of experimental timelines is provided in Fig. S1. Wild goldfish (4–7 cm total length) were collected from an urban stormwater pond in Hamilton during December 2024 using a 9.2 m straight seine net. Fish were transported to the University of Toronto Scarborough, Ontario, Canada, in aerated and insulated transportation tanks. Due to facility regulations, fish were quarantined for 2 months in a semi-natural outdoor mesocosm in ambient winter temperatures (6.03°C±4.10°C) in 167 liter static plastic

Biology Open

tanks (79×61 cm; 12 h: 12 h winter light:dark cycle) in groups of 22–25 individuals before being brought into the laboratory facility.

Upon being brought to the laboratory, we warmed fish ($N$=12) to average summer urban pond temperatures (22°C), at a slow rate of approximately 0.02°C/min. The 22°C summer average temperature was based on field recordings from six ponds provided by Fisheries and Oceans Canada; similar profiles for the nearby Coote's Paradise site can be found in Trueman (2021). After the warming period, fish were randomly divided into groups of 4–5 individuals into four replicate 76 liter tanks (60.96×30.48×40.64 cm), based on previous work suggesting 0.25 goldfish/l is the maximum density for maintenance of optimal growth rates (Jahedi et al., 2012). The photoperiod in our lab was a summer 14 h:10 h light:dark cycle. Tanks were enriched with slate rocks and sterilized plastic plants. Dissolved oxygen (DO) was recorded continuously and maintained at full saturation (90–100% DO). Individuals were fed *ad libitum* once a day, with commercial goldfish flake food (Nutrafin Max Goldfish flakes). Fecal matter and other debris were siphoned daily after each feeding to maintain water quality at healthy levels. Tank water was exchanged weekly (25%) and topped with commercial nitrifying bacteria (FritzZyme 7 Freshwater), per the manufacturer's specifications.

Fish were held in these common-garden acclimation conditions for a period of 3 weeks, consistent with the acclimation times used to achieve a 'stable state' in goldfish physiology studies (Fry and Hart, 1948a; Ferreira et al., 2014) and acclimation periods used for wild-caught fish (e.g. Christensen et al., 2021).

### Metabolic rate measurements

We estimated routine and maximum metabolic rates of fish indirectly at two acute test temperatures, 26°C and 30°C through oxygen consumption rate ($\dot{M}_{O2}$) using closed respirometry with a modification that allowed us to flush chambers with fresh water during a respirometer acclimation period (Nelson, 2016; Chabot et al., 2016; Raby et al., 2020). These temperatures were based on summer temperature logs onsite, showing daily maximum urban pond temperatures are approximately 26°C (Fisheries and Oceans Canada, pers. comm.); the 30°C test temperature was used to reflect an acute +4°C peak warming event. All fish were first tested for RMR and MMR at 26°C, allowed to rest for 10 days at common-garden conditions of 22°C, then tested again at 30°C (flowchart available in Fig. S1).

We chose a rest period of 10 days between trials to account for declining repeatability of metabolic rate measurements with time (White et al., 2012; Norin and Malte, 2011), while balancing a conservative rest period to recover from the cellular heat shock response of the first trial. Specifically, acute heat shocks do not typically lead to persistent whole-organism effects in ectotherms (Massey and Hutchings, 2021); when evaluated on the level of gene expression, studies have found transcripts return to normal within 8 h post-heat shock (salmonids; Lewis et al., 2016) and <5 days (killifish, following exposure to thermal cycling; Podrabsky and Somero, 2004). Testing occurred over multiple days starting at the same time (10:00) to standardize the timing for each test. Additional details, including the metabolic rate measurement checklist from Killen et al. (2021), are included in Table S1.

### RMR

We describe our metric herein as RMR rather than SMR because trials were conducted during the day for logistical reasons (facility regulations) and goldfish are generally a diurnal species (Herrera-Castillo et al., 2024). Fish were fasted for 48 h before measurements to reduce digestion-associated metabolism, after an initial pilot indicated fish still produced waste after 24 h.

The respirometry setup consisted of four glass respirometry chambers, made using 300 ml crystallization dishes sealed with 2-hole rubber stoppers. Two 0.64 cm inner diameter ball valves were adhered to the lid using silicone glue, allowing us to connect one tube to a water pump (Pawfly 400 GPH Submersible Water Pump, 1500 liters/h) set in a reservoir set to test temperatures, with the other diverting water back into the reservoir. We ran experiments using fresh, dechlorinated City of Scarborough tap water. As the recommended fish:chamber volume ratios for static respirometry vary from 1: 20 to 1:100 (Svendsen et al., 2016; Clark et al., 2013), we selected a respirometer volume of approximately 50 times that of an average fish in this

experiment (average mass of 5.76 g, 49 times the volume of the fish; range: 31–112 times fish volume). This size was large enough in diameter to allow the fish to comfortably maintain its position in the water column (Cech and Brauner, 2011) and did not exceed the maximum limit of 200 times the fish volume (Svendsen et al., 2016). A magnetic stir bar was placed in each chamber, and we used magnetic stir plates (Intllab Magnetic Stirrer MS-500) to induce gentle mixing, preventing oxygen stratification during measurement. During preliminary pilot trials, we observed that there was enough clearance between the stir bar and fish such that fish did not contact the stir bar, so we decided not to include a mesh divider to (i) increase the area available to the fish for maintaining its position in the water column; and, (ii) decrease possible surface area for microbial buildup during trials.

The respirometers were placed inside an opaque water bath set to test temperatures using a heater (Eheim Jager TruTemp Submersible 50 W Heater). Opaque plastic dividers were placed between respirometers such that fish could not see one another, nor the experimenters. Oxygen concentration was measured using contactless oxygen sensor spots, connected to a 4-channel oxygen meter with fiber optic cables (PyroScience GmbH, Aachen, Germany). Data were recorded in each chamber every second using the Oxygen Logger software (PyroScience). After each trial (RMR and MMR) was completed, fish were anesthetized using eugenol, the total length (1 mm) and mass (0.1 g) were measured. To ensure standard readings, each oxygen sensor was calibrated to 100% DO in water that was air-saturated with oxygen and to 0% DO in a sodium sulphite ($NaSO_2$) solution (30 g/l) daily, before trials. To prevent microbial build-up, equipment was cleaned between each trial using 70% ethanol.

In the morning, four randomly selected, fasted fish were individually placed in sealed respirometers in the water bath already set to test temperatures (i.e. fish experienced a change in temperature of 4°C and 8°C for the 26°C and 30°C trials, respectively, upon transfer). Each respirometer was gently flushed with oxygenated water from pumps for a period of 90 min to allow the fish to acclimate to the chamber and experimental temperature while reducing waste buildup inside the chamber (Steffensen, 1989; Svendsen et al., 2016; Paschke et al., 2018). This acclimation period was based on pilot trials in which chambers containing fish were manually flushed repeatedly to maintain normoxia while oxygen was being recorded over 3 h. A rolling regression, with intervals of 5 min, of measurement phases during the 3 h trials suggested minimum, stable respiration rates were achieved after ∼72 min (Fig. S2). After acclimation, pumps were shut off and valves were sealed, with care taken not to disturb fishes. Oxygen consumption was recorded until oxygen concentration was just below normoxia (i.e. 70–80% DO). At this point, valves were reopened, and the chambers were flushed with oxygenated water. The experimenters were careful not to disturb fish who were still undergoing recording. Fish were allowed to rest for at least 15 min before being removed in random order to undergo maximum metabolic rate trials.

### MMR

To determine the MMR, we used an exhaustive-chase protocol (Raby et al., 2020). One at a time and randomly with respect to the chamber, fish were taken from the respirometer and placed in a large rectangular arena (53×38×18 cm deep) set to test temperatures. Using clean, bare hands, fish were first encouraged to swim by agitating the water near the tail of the fish until exhaustion (i.e. they would no longer attempt to escape; Raby et al., 2020, Scheuffele et al., 2021). The fish were chased for an average of 10 min until they showed symptoms of exhaustion (i.e. no longer attempting to swim). Post-chasing, fish were immediately (i.e. within 15 s, giving the experimenter time to reopen the chamber) moved to the respirometers from the rectangular arena with minimal air exposure (Raby et al., 2020). The respirometer was then immediately sealed, and oxygen consumption measurements were taken in normoxia.

### Background respiration

While each fish underwent the chase protocol, their respective respirometers, still containing water that had contact with the fish, were sealed, and oxygen consumption measurements were taken as an estimate of background (microbial) respiration occurring alongside fish respiration in each chamber

to better reflect real measurement conditions than using a blank (sterile) chamber. The background respiration was measured on average for ∼10–15 min (i.e. as long as the individual was being chased) for each trial (per respirometer and fish).

## Data and statistical analyses

Background (microbial) respiration rate was estimated for each trial by calculating the total oxygen consumption during the background recording period (∼10–15 min) that occurred between RMR and MMR measurements. We corrected for background respiration by subtracting the slope of a best-fit regression line over the entire background recording period from both the RMR and MMR fish respiration values.

Whole-organism RMR was estimated using the quantile (q0.25) method, such that it was taken as the value that falls below 25% of all $MO_2$ values in normoxia (Chabot et al., 2016). The q0.25 method is expected to produce conservative estimates of RMR and is applicable to most fishes (Chabot et al., 2016). First, the $MO_2$ was calculated during 60 s RMR measurement intervals using the *calc_MO2* function in the *respirometry* package (Birk, 2024) iteratively on each 60 s recording interval:

$$MO_{2\,(60\,s)} = (\Delta O_2) \times V \times \Delta t^{-1}, \qquad (1)$$

where $\Delta O_2$ is the background-corrected change in oxygen concentration (mg $O_2$ $l^{-1}$) over time interval $\Delta t$ (60 s), multiplied by $V$, the respirometer's water volume minus the volume of the fish, (L). The average number of intervals used was 19 (range: 16–27).

To estimate whole-organism MMR, the *calc_MO2* function was used across the entire post-chase oxygen consumption trace in normoxia using:

$$MMR_{\Delta t} = (\Delta O_2) \times V \times \Delta t^{-1}, \qquad (2)$$

where $\Delta O_2$ is the background-corrected change in oxygen concentration (mg $O_2$ $l^{-1}$) over time interval $\Delta t$ (duration of all post-chase measurements in normoxia), multiplied by $V$, the respirometer's water volume minus the volume of the fish (l).

Whole-organism AS was estimated by subtracting the final RMR values from MMR values for each fish.

Individual Bayesian multilevel models were used to examine the effect of the treatment temperature (26°C and 30°C) on RMR, MMR, and AS. We used this approach because Bayesian models can better-describe variation in small datasets through the examination of the posterior effect sizes and their distributions (McNeish, 2016). However, we used uninformative (weak) default priors (McNeish, 2016), thus making our model results analogous to frequentist models. To account for differences in metabolic scaling between fish of different sizes, we included the fish mass as a covariate (Chabot et al., 2016). We also log-transformed outputs (RMR, MMR, and AS) and mass to account for nonlinear scaling between these variables. Tank replicate (1–3) was also included as a random effect to account for intra-tank variation within replicate tanks, but no inferences about these effects were made as random effect estimates are unreliable when there are fewer than five levels (Gomes, 2022). The models were run using the *brms* package in the R Computing Environment (Bürkner, 2018). Models used a Gaussian distribution with an identity link specified by *brms*. The model formulation was:

$$(Metabolic\ Traits)_{ij} = \beta_0 + \beta_1 T_i + \beta_2 M_i + u_{0j} + \varepsilon_{ij}, \qquad (3)$$

where 'Metabolic Traits' represent different models for log-transformed RMR, MMR, and AS; $\beta_0$ is the intercept, $\beta_1$ is the coefficient estimate for $T$, test temperature, $\beta_2$ is the coefficient estimate for the covariate, $M$ is the log-transformed fish mass, $u$ is the adjustment term for the random effect of tank replicate, and $\varepsilon$ is the error.

To aid in visualization, we also mass-standardized experimental data, such that RMR, MMR, and AS data have been standardized to the average body size of the fish in our experiment (5.76 g) using the methods outlined in Norin et al. (2015). Specifically, we used outputs from individual RMR, MMR, and AS models using mass as a covariate (i.e. coefficients) to calculate residuals for individual fish at each test temperature and added these to RMR, MMR, and AS values calculated for a standard 5.76 g fish.

Data, including trial date, fish masses, and whole-organism and standardized RMR, MMR, and AS are provided in Table S2.

## Acknowledgements
The authors thank the editors and reviewers for their valuable and thoughtful comments and suggestions that have helped improve the work, Mary Kate Fredricks for advice on animal husbandry, Samantha Burgh and Michael Nixon for assistance with animal husbandry, apparatus setup and/or cleanup, and The University of Toronto animal care and vivarium teams for daily monitoring of animal health and for providing access to the facility to conduct experimental trials and house our study animals.

## Competing interests
The authors declare no competing or financial interests.

## Author contributions
Conceptualization: N.A.N., N.E.M., M.D.M.; Data curation: N.A.N.; Formal analysis: M.D.M.; Funding acquisition: N.A.N., N.E.M., M.D.M.; Investigation: N.A.N., N.E.M., M.D.M.; Methodology: N.A.N., N.E.M., M.D.M.; Project administration: N.E.M., M.D.M.; Resources: N.E.M., M.D.M.; Software: M.D.M.; Supervision: N.E.M., M.D.M.; Validation: N.A.N., N.E.M., M.D.M.; Visualization: M.D.M.; Writing – original draft: N.A.N.; Writing – review & editing: N.A.N., N.E.M., M.D.M.

## Funding
This research was funded by a University of Toronto Excellence Award to N.A.N., Natural Sciences and Engineering Research Council of Canada Discovery Grant to N.E.M., and Liber Ero Foundation Scholarship to M.D.M. Open Access funding provided by University of Toronto Scarborough. Deposited in PMC for immediate release.

## Peer review history
The peer review history is available online at https://journals.biologists.com/bio/lookup/doi/10.1242/bio.062160.reviewer-comments.pdf

## Data and resource availability
All relevant data and details of resources can be found within the article and its supplementary information.

## Diversity and inclusion statement
N.A.N. and M.D.M. are racialized early-career scholars and we are grateful for the support of N.E.M., the University of Toronto Scarborough, and our funders.

## First Person
This article has an associated First Person interview with the first author of the paper.

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
