## [Peer Review File · Biology Open]

Invasive Goldfish (*Carassius auratus*) maintain aerobic scope across acute warm water temperatures

Nicholas Edward Mandrak; Melanie Duc Bo Massey and Nazeefa Arifina Nashrah

DOI: 10.1242/bio.062160

Editor: Lewis Halsey

Review timeline

Original submission:	13 May 2025
Editorial decision:	22 May 2025
Resubmission:	10 July 2025
Editorial decision:	17 July 2025
First revision received:	28 July 2025
Accepted:	29 July 2025

Original submission

First decision letter

MS ID#: bio.062066

MS TITLE: Invasive Goldfish (*Carassius auratus*) maintain aerobic scope across acute warm water temperatures

AUTHORS: Nazeefa Arifina Nashrah; Nicholas Edward Mandrak; Melanie Duc Bo Massey

Dear Dr Hirata,

I am writing to let you know that I have now reached a decision on the above manuscript. I am afraid that, after careful consideration, I feel that it cannot currently be accepted for publication in Biology Open.

The reviewer reports are shown at the bottom of this email or can be accessed, together with a copy of this decision letter, by going to:

As you will see, the reviewers raise a number of substantial criticisms that prevent me from accepting your paper for publication. For example, both have major concerns about the experimental design, and reviewer 2 questions whether your study surpasses a related study from 1948.

I realise that this is disappointing news, and we understand the frustration that you must feel. However, I am sure that you appreciate that the conclusions of your research must be seen by the wider community to be fully supported by the data. On this occasion, I have decided that this is not the case.

I do hope you find the reviewer comments helpful in allowing you to revise the manuscript for successful submission to Biology Open or elsewhere.

Comments from the Reviewers:

Reviewer 1: This study investigated the temperature-dependence of aerobic metabolic scope of (invasive) goldfish. The salient finding is that aerobic scope is maintained at high temperatures, i.e. is about the same at 26 and 30°C. The authors argue and discuss that this maintained aerobic scope is likely behind the ability of the species to occupy and invade a broad range of habitats around the world. The narrative is heavily directed towards aerobic scope being an established fitness proxy within the oxygen- and capacity-limited thermal tolerance (OCLTT) framework.

While the manuscript is generally well written, I unfortunately think there are critical issues with the study and storyline that complicate any robust conclusions. I have outlined these below. I regret that I cannot be more positive, but I hope the authors can use my comments when designing future studies.

General comments:

OCLTT and the ecological relevance of aerobic scope in a climate change context are heavily debated (e.g. see Jutfelt et al. 2018). The authors do not mention or acknowledge this debate at all, which ought to be done. The authors also claim that the temperature dependence of aerobic scope in goldfish is unknown, but they have overlooked the arguably most seminal study in the context of temperature dependence of aerobic scope - the study by Fry & Hart (1948) on goldfish, from the authors' own university. Fry & Hart measured both maximal (active) and standard metabolic rates, and thus aerobic scope, at 6-7 temperatures from 5 to 35°C, and show that aerobic scope plateaus between 25 and 30°C. The present study also finds that aerobic scope is the same at 26 and 30°C but, as the authors did not measure aerobic scope at additional temperatures, they cannot say if aerobic scope was higher or lower at higher or lower temperatures than the two investigated here. Fry & Hart thus still stands as the more comprehensive study despite being conducted nearly 80 years ago.

I also think that the methodology of the present study prevents conclusions in the context of climate change and heat waves, as the authors set it within:

First, the fish were sampled in a small pond in the Canadian winter (December 2024) and subsequently held for two months in outdoor mesocosms at ambient but undisclosed temperature (but presumably close to freezing), then (only) given three weeks to acclimate to a comparably very high temperature of 22°C in the lab, after which it sounds like the fish were put directly into the 26 or 30°C water that their oxygen uptake rates were measured at as a proxy for metabolic rate. None of these temperature exposure scenarios resemble climate warming or heat waves, so the measurements of aerobic scope can most likely not say anything about the goldfish' actual ability to perform at high temperatures in a meaningful ecological context. Additionally, the authors imply on Page 11, line 40-41 that the fish were in overwintering hibernation after they caught them (i.e. while in the outdoor mesocosms for two months during winter). If this was actually the case, it further questions any ecological relevance of warming and measuring metabolic rates of the fish from a state of hibernation to 22 and then 26 or 30°C within 4 weeks.

Second, metabolic rates was measured on the same fish at 26 and 30°C, but non-randomized, such that all individuals were measured at 26°C first and then at 30°C a week later. As the initial exposure to 26°C is likely to have induced some form of thermal response at the cellular and molecular levels (as the authors themselves discuss at the end of Page 11 and start of page 12), subsequent measurements at 30°C are likely to reflect such thermal compensation responses and thus not independent of the previous treatment. I guess it could be argued that this resembles heat waves more than randomized exposure to 26 and 30°C, but it is not ideal from an experimental design perspective.

Third, aerobic scope is defined as maximum minus standard metabolic rate, but the authors measured routine metabolic rate. That is, some level of metabolic rate above standard. As routine metabolic rate likely includes some measure of activity, which may change with temperature, it is difficult to say how these measurements of routine metabolic rate affected the calculated aerobic scope. The technique to measure routine metabolic rate is somewhat odd too. It has long been accepted that intermittent-flow respirometry is the preferred method for estimating metabolic rates from rates of oxygen uptake (e.g. Steffensen 1989; Killen et al. 2021). The present study

applied some technique intermediate between closed respirometry and intermittent-flow respirometry, in which the fish were left in the respirometry chambers for 1.5 h with flushing of new water into the chambers, after which the chambers were manually sealed by turning off a valve located on top of the chambers, presumably disturbing the fish in the process. The drop in oxygen concentration from fish respiration was then recorded until oxygen level reached 80% air-saturation. This must have happened at different times for different individuals (of different body sizes), again causing disturbance to the remaining individuals when those reaching 80% had the top valves reopened for flushing the chambers with fresh, aerated water. For calculations of metabolic rate, the closed period between ~100 and 80% air-saturation were then dissected into several 1-min chunks for which the oxygen uptake rate of the fish was calculated. The 25th quantile of these oxygen uptake rates was taken as routine metabolic rate. There is no description of how long it took the fish to go from 100 to 80% air-saturation, but presumably not very long at 26 or 30°C in respirometry chambers that were appropriately sized at 30-50 times the volume of the fish.

I strongly recommend that the authors adopt the intermittent-flow respirometry technique in future work. Killen et al. (2021) is a great resource for this methodology. The authors already have the most expensive piece of equipment (the FireSting oxygen meter), so all the intermittent-flow technique would require is another set of (inexpensive) aquarium pumps and a reliable relay (timer) to control the flush pumps (e.g. something like this: <https://export.rsdelivers.com/product/rs-pro/rs-pro-panel-mount-timer-relay-110-240v-ac-2-01s/8966872>, mounted in a cabinet).

I do not think the statistical analyses the authors apply are appropriate. First, they express metabolic rates as mass-specific (text and Fig. 1). However, simply dividing by body mass does not remove the effect of body mass, unless metabolic rate scales linearly with body mass. This is not necessarily a problem, if body masses were not significantly different between treatment groups, but there is no information about this. These data ought to be included. The authors then complement these mass-specific metabolic rates with Bayesian mixed-effects models of whole-animal metabolic rates as a function of both temperature and body mass. While I personally think Bayesian modelling is overkill here, the authors model the relationships between metabolic rate and both temperature and body mass as linear, which they most likely are not. The relationship between metabolic rate and body mass usually follows a power function, while for temperature it is usually exponential. The authors thus ought to log-transform metabolic rate and body mass, but not temperature, in their models to account for this. I think the mix of these mass-specific and modelled data in the Results section is a bit confusing and could need some streamlining. It is unclear which results are most relevant/correct and why both are needed. I also think the presentation of posterior distributions in Fig. 2 is unnecessarily complicated and difficult to understand for most readers. Are they necessary? If there are no differences in mean body mass between the two temperature groups, presenting the mass-specific or whole-animal metabolic rates should suffice. If there are differences, metabolic rates could be standardized to a common (mean) body mass by adding the residuals from the model to the predicted metabolic rate for a fish of that common body mass, at each temperature, if the relationships between metabolic rate and body mass is significant.

Finally, referencing could be tightened up throughout. There are several places where the cited literature does not support the statement being made. For example:

Page 3, lines 22-24: Christensen et al. (2021) does not support that "[aerobic scope] representing the rate at which resources from the environment are converted into energy."

Page 3, lines 33-37: Clark et al. (2013) does not support that "animals should have optimal fitness when living within temperature ranges where aerobic scope is maximized."

Page 4, lines 46-50: The relevance of Auer et al. (2015a) is unclear in the statement "To understand how invasions may change under climate-change scenarios, it is important to assess changes in ecologically relevant aerobic performance metrics of invaders."

Page 5, lines 26-28: Unclear what Christensen et al. (2021) has to do with the statement that "Fish were then acclimated to baseline summer conditions (22°C; 14 h: 10 h Summer light: dark cycle) for three weeks."

Page 6, line 25: Here, the authors refer to an initial pilot experiment they made for their study with reference to a study by Frisk et al. (2013) from more than a decade ago. Unclear what this has to do with their pilot.

Page 6, lines 49-51: The authors state here that "To determine the maximum metabolic rate, we used an exhaustive-chase protocol (Raby et al., 2020)." However, Raby et al. (2020) shows that the chase protocol is not the best method to elicit maximum metabolic rate.

Page 11, lines 45-51: Ferreira et al. (2014) is not in crucian carp, as indicated.

Specific comments.

Page 5, lines 12-18: Please add what the water temperature was at the time of fish collection and/or during the 2-months holding in the outdoor mesocosms.

Page 5, lines 26-28: How were fish exposed to 22°C Directly (acutely) from the outdoor mesocosms, or gradually warmed at a specified rate? This is important information that must be included.

Page 5, lines 55-56: How was the stir bar 'affixed to each chamber'? What prevented the fish from touching the stir bar?

Page 6, lines 11-12: The water would not have been 'oxygen-saturated' but rather air-saturated with oxygen. 'Oxygen-saturated' implies that the water was saturated with pure oxygen.

Page 6, lines 53-59: Please add for how long the fish were chased. On the next page, the authors state that 10 min recordings of background respiration were taken while the fish were being chased, so presumably the fish were chased for at least 10 min each. This is very long for small 3-9 gram fish. What is the reasoning behind these long chase times?

Page 7, lines 4-10: It says here that "Post-chasing, fish were immediately moved to respirometers with minimal air exposure, which has been shown to improve the accuracy of MMR measurements using the chase protocol (Raby et al., 2020). The respirometer was then immediately sealed and oxygen consumption measurements were taken in normoxia." If recordings of background respiration was ongoing while the fish were being chased, the respirometry chambers must have been sealed and needed to also be opened before the fish could be transferred there. How long did it take to transfer the fish from cessation of chasing? And where was the fish while chambers were being opened?

Page 7, line 13: This section is about measuring background respiration, not correcting for it. That comes later.

Page 7, lines 28-30: How were fish exposed to 26 and 30°C? Directly (acutely) from 22°C, or gradually warmed at a specified rate? This is important information that must be included.

Page 8, Eq. 1 and Eq. 2: There is no definition of 'm' (it is obviously body mass but ought to be defined).

Page 9, Eq. 3: See general comment above about log-transformation. Also, the constants beta 1 and beta 2 are not defined (whereas beta 0 is).

Page 9, line 17: It is unclear how the mass-adjustment becomes 'per kg' given that MO₂ values (cf. Eqs. 1 and 2) are divided by body mass, and body mass is in grams, not kg, according to the supplementary data table.

Page 9, lines 17-21: The authors present differences in metabolic rates between the two experimental temperatures here. It would be more informative to present values for each temperature separately. It is also unclear what is meant by "differences in the average of the mass-

adjusted aerobic scope at 26°C and 30°C was negligible (179.5 mgO₂kg⁻¹h⁻¹ and 217.4 mgO₂kg⁻¹h⁻¹ for 26°C and 30°C respectively)." There is only one aerobic scope measurement at each temperature, so what are these 'differences in the average of the mass-adjusted aerobic scope'?

Page 9, lines 54-55: Aerobic scope is not a "widely accepted proxy of fitness". See Clark et al. (2013) and Jutfelt et al. (2018).

Page 10, line 8: Raby et al. (2016) is not in the reference list.

Page 10, lines 8-10: The "ecological relevance of aerobic scope" is also not set in stone. Please consider the debate about aerobic scope and OCLTT (again, see Jutfelt et al. 2018 for summary).

Page 11, lines 19-23: The authors state here that "if Goldfish are acclimated to warmer conditions (i.e. throughout summer), it might be possible that the aerobic scope would further increase at the higher test temperature." Is it not more likely that aerobic scope would decrease with longer acclimation time, as is generally seen when ectotherms thermally compensate / acclimate (e.g. Norin et al. 2014)?

Page 22, Fig. 1. It would be good to calculate and add Q10s for the different metabolic rates. It would perhaps also make sense to plot RMR and MMR in the same panel, which would make it easier to visualize aerobic scope.

Cited literature (that is not already in reference list):

Fry FEJ & Hart JS (1948) The relation of temperature to oxygen consumption in the goldfish. *The Biological Bulletin* 94, 66-77.

Jutfelt F, Norin T, Ern R, Overgaard J, Wang T, McKenzie DJ, Lefevre S, Nilsson GE, Metcalfe NB, Hickey AJR, Brijs J, Speers-Roesch B, Roche DG, Gamperl AK, Raby DG, Morgan R, Esbaugh AJ, Gräns A, Axelsson M, Ekström A, Sandblom E, Binning SA, Hicks JW, Seebacher F, Jørgensen C, Killen SS, Schulte PM & Clark TD (2018) Oxygen- and capacity-limited thermal tolerance: blurring ecology and physiology. *Journal of Experimental Biology* 221, jeb169615.

Killen SS, Christensen EAF, Cortese D, Závorka L, Norin T, Cotgrove L, Crespel A, Munson A, Nati JJH, Papatheodoulou M & McKenzie DJ (2021) Guidelines for reporting methods to estimate metabolic rates by aquatic intermittent-flow respirometry. *Journal of Experimental Biology* 224, jeb242522.

Steffensen JF (1989) Some errors in respirometry of aquatic breathers: how to avoid and correct for them. *Fish Physiology and Biochemistry* 6, 49-59.

Reviewer 2: This study addresses a timely and ecologically relevant question regarding the physiological capacity of invasive goldfish to tolerate high water temperatures, with potential implications for their spread under climate change. The authors use aerobic scope as a physiological proxy to infer potential fitness at two elevated temperatures, building a narrative around the ecological risk posed by thermally tolerant invaders. However, there are several conceptual and methodological issues that weaken the study and conclusions, particularly around the interpretation of aerobic scope, experimental design, and data analysis. These should be addressed for the manuscript before being publishable.

- The manuscript refers to aerobic scope as a proxy for energy able to be allocated for reproduction, growth, and other fitness-related traits. However, this claim is often stated more than it is actually observed, and references would help support the argument that there is a link between aerobic scope and fitness. Also, aerobic scope represents the difference between maximum and standard (or occasionally routine) metabolic rate and reflects capacity, not actual energy allocation or utilization.

- The use of RMR instead of SMR to estimate aerobic scope is problematic. RMR includes spontaneous activity and can differ substantially from SMR, especially if activity levels change across temperatures. If fish were less active at 30°C, for instance (as may occur if they are truly aerobically compromised), the RMR might decrease despite elevated metabolic demands, artificially inflating aerobic scope estimates. The manuscript should either justify the use of RMR over SMR or reframe the interpretation of the data accordingly.
 - The experimental design introduces a serious temporal confound: all fish were measured at 26°C first, followed by 30°C a week later. It is therefore unclear whether the observed differences between temperatures are due to warming or extended lab acclimation. A counterbalanced or randomized temperature order would have been preferable. As is, any interpretation of temperature effects must be cautious and this caveat should be emphasized in both methods and discussion.
 - The use of a short acclimation period (stated as "at least 1 week") prior to the 30°C trial is ambiguous. It's unclear whether all fish were given the same duration of acclimation, and whether this is sufficient to assess steady-state physiological responses.
 - Measurement of SMR/RMR lacks detail. The authors cite a 1.5-hour acclimation period in the respirometer, yet most literature suggests that SMR (or even stable RMR) is only achieved after much longer habituation periods (typically 5-24 hours). The relatively short chamber time may result in elevated MO₂ readings due to stress or activity, underestimating AS. This limitation should be acknowledged clearly.
 - The paper should clarify whether closed or intermittent-flow respirometry was used. While the text implies closed respirometry, the protocols and calculations would benefit from further clarification.
 - The distinction between mass-specific and mass-adjusted metabolic rates is blurred. The manuscript describes values as "mass-adjusted," yet it appears the rates are simply expressed per kg (by dividing RMR by mass), without statistical correction (mass-specific metabolic rate decreases with body mass). If linear models were used to adjust for mass, then the estimates reported should be model outputs or adjusted residuals.
 - The quantile method (q0.25) for RMR is fine in principle, but the authors do not report the number of slopes used or whether these were independent. If slopes are tightly clustered in time (which I think they must have been because they seem to be using closed respirometry), the quantile estimate may be artificially narrow due to temporal autocorrelation.
 - The manuscript states that "little is known" about temperature effects on aerobic scope in goldfish. This is inaccurate as several prior studies have examined these relationships, including: E.G. Ferreira, E. O., Anttila, K., & Farrell, A. P. (2014). Thermal optima and tolerance in the eurythermic goldfish (*Carassius auratus*): relationships between whole-animal aerobic capacity and maximum heart rate. *Physiological and Biochemical Zoology*, 87(5), 599-611. Pang, X., Cao, Z. D., & Fu, S. J. (2011). The effects of temperature on metabolic interaction between digestion and locomotion in juveniles of three cyprinid fish (*Carassius auratus*, *Cyprinus carpio* and *Spinibarbus sinensis*). *Comparative Biochemistry and Physiology Part A: Molecular & Integrative Physiology*, 159(3), 253-260.
- The introduction and discussion should be revised to more accurately reflect the existing literature.

Reviewer's Responses to Questions

Experimental quality

Does each figure have the proper controls?

If 'No', please indicate reasons in Comments for Author box below.

Reviewer #1:

- Yes

Reviewer #2:

- Yes

Were the data analyzed using appropriate statistical tests?

If 'No', please indicate reasons in Comments for Author box below.

Reviewer #1:

- No

Reviewer #2:

- Yes

Reproducibility

Were experiments performed using adequate number of biological replicates?

If 'No', please indicate reasons in Comments for Author box below.

Reviewer #1:

- Yes

Reviewer #2:

- Yes

Does the methods section provide sufficient detail to permit reproducibility?

If 'No', please indicate reasons in Comments for Author box below.

Reviewer #1:

- No

Reviewer #2:

- No

Completeness

Are the manuscript's conclusions supported by the data?

If 'No', please indicate reasons in Comments for Author box below.

Reviewer #1:

- No

Reviewer #2:

- No

Scholarship

Do the authors cite and discuss the merits of data that would argue for and against their conclusion?

If 'No', please indicate reasons in Comments for Author box below.

Reviewer #1:

- No

Reviewer #2:

- No

Does the manuscript title & abstract accurately reflect the contents of the manuscript, without hyperbole?

If 'No', please indicate reasons in Comments for Author box below.

Reviewer #1:

- Yes

Reviewer #2:

- Yes

Author response to reviewers' comments

Reviewer 1:

This study investigated the temperature-dependence of aerobic metabolic scope of (invasive) goldfish. The salient finding is that aerobic scope is maintained at high temperatures, i.e. is about the same at 26 and 30°C. The authors argue and discuss that this maintained aerobic scope is likely behind the ability of the species to occupy and invade a broad range of habitats around the world. The narrative is heavily directed towards aerobic scope being an established fitness proxy within the oxygen- and capacity-limited thermal tolerance (OCLTT) framework.

While the manuscript is generally well written, I unfortunately think there are critical issues with the study and storyline that complicate any robust conclusions. I have outlined these below. I regret that I cannot be more positive, but I hope the authors can use my comments when designing future studies.

We are grateful to the Reviewer for their comprehensive constructive criticism. The suggestions you have given us are very helpful for improving both this paper and our methods going forward. We are also very appreciative that you took the time to share relevant literature with us. We've responded to your comments in red, noting where we have made changes in the manuscript.

General comments:

OCLTT and the ecological relevance of aerobic scope in a climate change context are heavily debated (e.g. see Jutfelt et al. 2018). The authors do not mention or acknowledge this debate at all, which ought to be done. The authors also claim that the temperature dependence of aerobic scope in goldfish is unknown, but they have overlooked the arguably most seminal study in the context of temperature dependence of aerobic scope - the study by Fry & Hart (1948) on goldfish, from the authors' own university. Fry & Hart measured both maximal (active) and standard metabolic rates, and thus aerobic scope, at 6-7 temperatures from 5 to 35°C, and show that aerobic scope plateaus between 25 and 30°C. The present study also finds that aerobic scope is the same at 26 and 30°C but, as the authors did not measure aerobic scope at additional temperatures, they cannot say if aerobic scope was higher or lower at higher or lower temperatures than the two investigated here. Fry & Hart thus still stands as the more comprehensive study despite being conducted nearly 80 years ago.

We agree that the work of Fry & Hart (1948) is comprehensive and we regret not including this foundational work while writing our original manuscript. We would like to point out notable differences between their study and ours. First, Fry & Hart (1948) [as well as many other existing

Goldfish thermal metabolism studies, e.g. Kanungo & Prosser (1959), Ferreira et al., (2014)] used commercial strains of Goldfish, acquired from the aquaculture industry or pet stores. Our Goldfish were wild-caught invaders from a local site. This is significant because (1) there *may* be plastic/genetic adaptation that has occurred in our wild-caught study fish that lead to differential responses to temperature than what has previously been studied, and, importantly, (2) there are local management/conservation implications that can only be known by directly studying local populations. With respect to this point, we have significantly reworked the introduction to highlight the invasion biology context of our study. However, acknowledging that we failed to initially incorporate existing relevant literature, we have also added in a review of what is known about Goldfish metabolism from foundational studies. A second point is that in Fry & Hart (1948), fish were *tested* at their respective *acclimation* temperatures (e.g., a fish that was tested at 20°C was also acclimated at 20°C). There was no common-garden acclimation. Therefore, we do not believe that this is a full representation of the ecophysiology of this species, which will experience variation in ambient temperatures prior to exposure to ecologically relevant warm periods - as a cohort.

I also think that the methodology of the present study prevents conclusions in the context of climate change and heat waves, as the authors set it within:

First, the fish were sampled in a small pond in the Canadian winter (December 2024) and subsequently held for two months in outdoor mesocosms at ambient but undisclosed temperature

(but presumably close to freezing), then (only) given three weeks to acclimate to a comparably very high temperature of 22°C in the lab, after which it sounds like the fish were put directly into the 26 or 30°C water that their oxygen uptake rates were measured at as a proxy for metabolic rate.

Although we acknowledge that we did not provide this information initially, the temperature acclimation and exposures actually do reflect the natural mean summer temperatures and daily maxima experienced in our study sites (urban ponds). We described this in more detail in our methods. We have added that the average temperature of the ambient ‘quarantine’ mesocosms was 6.03°C +/- 4.10°C and that fish were adjusted to average summer pond temperature (22°C) in lab at a rate of 0.02°C per minute (over approximately 8 hours). We then continued to acclimate fish at 22°C for 3 weeks. We acknowledge that this is a fairly short timeframe, but were limited as our undergraduate student author had to graduate. Further, 3 weeks of common-garden acclimation is not uncommon in ecophysiology studies using wild-caught fish (e.g. Christensen et al., 2021) or even in the Goldfish physiology literature that you suggested we review - indeed, Fry & Hart (1948) suggested ~20 days as the maximum duration of acclimation of Goldfish in their own experiment, Ferreira et al. (2014) used 3 weeks, Herrera-Castillo et al. (2024) used 14 days. We have added a brief explanation of this in our methods.

None of these temperature exposure scenarios resemble climate warming or heat waves, so the measurements of aerobic scope can most likely not say anything about the goldfish' actual ability to perform at high temperatures in a meaningful ecological context.

These temperatures were based on real data from urban ponds (shared by a contact at Fisheries and Oceans Canada, which we have now cited as a personal communication). We apologize that this was unclear and have explained it in more detail in the methods. Basically, these ponds, on average, are about 22°C and maximum average daily temperatures in summer are about 26°C. The 30°C represents a +4 C warming scenario.

Additionally, the authors imply on Page 11, line 40-41 that the fish were in overwintering hibernation after they caught them (i.e. while in the outdoor mesocosms for two months during winter). If this was actually the case, it further questions any ecological relevance of warming and measuring metabolic rates of the fish from a state of hibernation to 22 and then 26 or 30°C within 4 weeks.

We agree this is not ideal and have identified it as a limitation of our study in the Discussion.

Second, metabolic rates was measured on the same fish at 26 and 30°C, but non-randomized, such that all individuals were measured at 26°C first and then at 30°C a week later. As the initial exposure to 26°C is likely to have induced some form of thermal response at the cellular and molecular levels (as the authors themselves discuss at the end of Page 11 and start of page 12), subsequent measurements at 30°C are likely to reflect such thermal compensation responses and thus not independent of the previous treatment. I guess it could be argued that this resembles heat waves more than randomized exposure to 26 and 30°C, but it is not ideal from an experimental design perspective.

We agree, as Reviewer 2 have also mentioned, that this is something we cannot change. However, it might be realistic to consider that short pond heat shocks to a lower temperature (e.g. 26°C) might occur prior to warmer heat shocks (e.g., 30°C) as summers progress. We also explain in greater detail the assumption that 1 week is likely sufficient to recover from any potential heat shock that occurred over the previous trial period in our methods. We have, however, noted this as a limitation in the Discussion.

Third, aerobic scope is defined as maximum minus standard metabolic rate, but the authors measured routine metabolic rate. That is, some level of metabolic rate above standard. As routine metabolic rate likely includes some measure of activity, which may change with temperature, it is difficult to say how these measurements of routine metabolic rate affected the calculated aerobic scope.

We initially chose to consider our measurements “RMR” rather than “SMR” in order to be

transparent about our data, mainly due to the fact that we did not measure MO_2 during a strict “rest” period for our fish (we have observed that our study goldfish are inactive during lights off in the evening, with minimal movement or signs of awareness if we quietly move around the lab; goldfish are also known to be diurnal). Basically, we fell into the group that Chabot et al. (2016) mentioned: “...many authors prefer the term routine MR (RMR), which includes a minor cost of activity...”

Because we measured fish in the morning (from 10-11 am) due to facility regulations prohibiting nighttime experimentation, we assumed they were not “at rest” using the strictest definition of “rest”.

Ultimately, we did not want to mislead readers into assuming that we measured true SMR in the strictest sense of the term, although, to our knowledge, most fish metabolism studies do not actually ensure total inactivity/rest when SMR is measured. At the same time, we did use the quantile method to estimate RMR, which can help minimize MO_2 variation caused by ‘minor costs of activity’. In light of your comment, we have made explicit the definition of RMR in our methods with reference to the inconsistent use of the terms RMR/SMR.

The technique to measure routine metabolic rate is somewhat odd too. It has long been accepted that intermittent-flow respirometry is the preferred method for estimating metabolic rates from rates of oxygen uptake (e.g. Steffensen 1989; Killen et al. 2021).

We only had the materials required for static respirometry, but we sought to improve it by flushing chambers during the acclimation period. We are hoping to update our setup to align with best-practices going forward and thank you for the methodological suggestions.

The present study applied some technique intermediate between closed respirometry and intermittent-flow respirometry, in which the fish were left in the respirometry chambers for 1.5 h with flushing of new water into the chambers, after which the chambers were manually sealed by turning off a valve located on top of the chambers, presumably disturbing the fish in the process.

We cannot exclude this possibility. Without access to peristaltic pumps, we used a frugal design, in an attempt to minimize nitrogenous waste buildup and oxygen consumption during acclimation. Care was taken to avoid moving the respirometers when sealing valves. However, with intermittent-flow respirometry, the pumps coming back on is also a possible source of disturbance to the fish and water/microbes in the tubing and pump system add additional measurement error.

The drop in oxygen concentration from fish respiration was then recorded until oxygen level reached 80% air-saturation. This must have happened at different times for different individuals (of different body sizes), again causing disturbance to the remaining individuals when those reaching 80% had the top valves reopened for flushing the chambers with fresh, aerated water.

We cannot exclude this possibility, but we were careful not to disturb fish (visually by hovering or mechanically). We noted this in our methods.

For calculations of metabolic rate, the closed period between ~100 and 80% air-saturation were then dissected into several 1-min chunks for which the oxygen uptake rate of the fish was calculated. The 25th quantile of these oxygen uptake rates was taken as routine metabolic rate. There is no description of how long it took the fish to go from 100 to 80% air-saturation, but presumably not very long at 26 or 30°C in respirometry chambers that were appropriately sized at 30-50 times the volume of the fish.

You are correct in your interpretation of the analysis, and we have reported the number of 1-min time intervals used for our quantile estimations in the methods. Because the number of intervals was quite low (average: 19), we used the highest quantile (0.25) reported for use in fish in Chabot & Farrell (2016), which should have more conservatively estimated RMR than a lower quantile.

I strongly recommend that the authors adopt the intermittent-flow respirometry technique in future work. Killen et al. (2021) is a great resource for this methodology. The authors already have the most expensive piece of equipment (the FireSting oxygen meter), so all the

intermittent-flow technique would require is another set of (inexpensive) aquarium pumps and a reliable relay (timer) to control the flush pumps (e.g. something like this:

<https://export.rsdelivers.com/product/rs-pro/rs-pro-panel-mount-timer-relay-110-240v-ac-2-01s/8966872>, mounted in a cabinet).

We thank you for the excellent resource from Killen et al. and the suggestion to use a relay timer. We struggled with equipment limitations during this experiment but are working on improving our methods. We have reviewed Killen et al. 2021 and have added details to our methods based on the checklist and included a table in the supplementary data that details all the information that should be reported.

I do not think the statistical analyses the authors apply are appropriate. First, they express metabolic rates as mass-specific (text and Fig. 1). However, simply dividing by body mass does not remove the effect of body mass, unless metabolic rate scales linearly with body mass. This is not necessarily a problem, if body masses were not significantly different between treatment groups, but there is no information about this. These data ought to be included.

These data were included in our data file but since analysis of mass is not the focal goal of this piece we did not illustrate it in figures. We did make mass-standardization (to a 5.76 g fish) as you suggested below for Fig 1.

The authors then complement these mass-specific metabolic rates with Bayesian mixed-effects models of whole-animal metabolic rates as a function of both temperature and body mass. While I personally think Bayesian modelling is overkill here...

Our Bayesian models are analogous to frequentist models, we prefer them because we feel that they are more descriptive with respect to effect magnitude, direction, and variability of effects.

... the authors model the relationships between metabolic rate and both temperature and body mass as linear, which they most likely are not. The relationship between metabolic rate and body mass usually follows a power function, while for temperature it is usually exponential. The authors thus ought to log-transform metabolic rate and body mass, but not temperature, in their models to account for this.

We thank you for this suggestion and we have rerun models with log-transformed mass and MO₂ values to account for possible non-linearity. We have added an additional note in our methods to explain that this changes the interpretation of model results to a multiplicative scale (which is often considered more easily interpreted, e.g. change in condition leads to % increase/decrease in Y). We note that our results are not qualitatively changed (i.e., there are still moderate increases in MO₂ for SMR/MMR from 26->30C, but a “negligible” (zero) effect of the temperature changes for AS.)

I think the mix of these mass-specific and modelled data in the Results section is a bit confusing and could need some streamlining. It is unclear which results are most relevant/correct and why both are needed.

We wanted to show our ‘actual’ data in Fig. 1 but describe model results in Fig. 2 (tabular format is not strictly necessary, as it would be in frequentist statistics). For reference, we used the standardization methods described in detail by Norin et al. (2015; <https://besjournals.onlinelibrary.wiley.com/doi/full/10.1111/1365-2435.12503>).

I also think the presentation of posterior distributions in Fig. 2 is unnecessarily complicated and difficult to understand for most readers.

We find the visual effect sizes to be more descriptive of the data than a table of model outputs. We have decided to keep the visual output, but we have simplified it to include both the effect size distribution and interval plots (with dots for medians and line thicknesses illustrating 90 and 50% uncertainty intervals) rather than distributions.

Are they necessary? If there are no differences in mean body mass between the two temperature groups, presenting the mass-specific or whole-animal metabolic rates should suffice. If there are differences, metabolic rates could be standardized to a common (mean) body mass by adding the residuals from the model to the predicted metabolic rate for a fish of that common body mass, at each temperature, if the relationships between metabolic rate and body mass is significant.

We thank you for this suggestion. In response to your comment and a similar comment from Reviewer 2, we have recreated Fig. 1, such that reported metabolic values are standardized to the mass of an average fish in this experiment (5.76 g) using residuals from our models in which mass was treated as a covariate and have clarified this in the text of the methods.

Finally, referencing could be tightened up throughout. There are several places where the cited literature does not support the statement being made. For example:

Page 3, lines 22-24: Christensen et al. (2021) does not support that "[aerobic scope] representing the rate at which resources from the environment are converted into energy."

We have removed this reference and have rewritten the definition of the aerobic scope in our introduction.

Page 3, lines 33-37: Clark et al. (2013) does not support that "animals should have optimal fitness when living within temperature ranges where aerobic scope is maximized."

We have removed this reference and have de-emphasized suggestions about the ecological relevance of AS.

Page 4, lines 46-50: The relevance of Auer et al. (2015a) is unclear in the statement "To understand how invasions may change under climate-change scenarios, it is important to assess changes in ecologically relevant aerobic performance metrics of invaders."

We have changed this citation, which was placed there in error.

Page 5, lines 26-28: Unclear what Christensen et al. (2021) has to do with the statement that "Fish were then acclimated to baseline summer conditions (22°C; 14 h: 10 h Summer light: dark cycle) for three weeks."

We have removed this citation - Christensen et al acclimated their fish for 3 weeks (a common baseline/common-garden acclimation period for wild-caught fish).

Page 6, line 25: Here, the authors refer to an initial pilot experiment they made for their study with reference to a study by Frisk et al. (2013) from more than a decade ago. Unclear what this has to do with their pilot.

We have removed this citation.

Page 6, lines 49-51: The authors state here that "To determine the maximum metabolic rate, we used an exhaustive-chase protocol (Raby et al., 2020)." However, Raby et al. (2020) shows that the chase protocol is not the best method to elicit maximum metabolic rate.

Unfortunately, we did not have an appropriately sized swim flume to induce sustained swimming exhaustion. We did acknowledge this limitation in the discussion and mentioned that Raby et al. suggest this will lead to under-estimation of MMR, presumably evenly across the 2 test temperatures and is, therefore, a conservative limitation with respect to our results.

Page 11, lines 45-51: Ferreira et al. (2014) is not in crucian carp, as indicated.

This has been corrected with the proper citations.

Specific comments.

Page 5, lines 12-18: Please add what the water temperature was at the time of fish collection and/or during the 2-months holding in the outdoor mesocosms.

The information has been added; the average temperature in the preceding 2-month period was $6.03^{\circ}\text{C} \pm 4.10^{\circ}\text{C}$ on average.

Page 5, lines 26-28: How were fish exposed to 22°C Directly (acutely) from the outdoor mesocosms, or gradually warmed at a specified rate? This is important information that must be included.

The fish were gradually warmed at $0.2^{\circ}\text{C} / \text{min}$ with constant monitoring for any signs of stress

Page 5, lines 55-56: How was the stir bar 'affixed to each chamber'? What prevented the fish from touching the stir bar?

Stir bars were set to a low speed, and we did not observe the fish touching the stir bar during pilot or regular trials (there was enough clearance). We originally planned to include a mesh dome (i.e. a sink strainer) on top, but this reduced the amount of room each fish had in the chamber. Without increasing the chamber size (which would have made the volume:fish ratio too large), this setup appeared to work reasonably. We have explained this more clearly in our methods.

Page 6, lines 11-12: The water would not have been 'oxygen-saturated' but rather air-saturated with oxygen. 'Oxygen-saturated' implies that the water was saturated with pure oxygen.

We have updated the line accordingly

Page 6, lines 53-59: Please add for how long the fish were chased. On the next page, the authors state that 10 min recordings of background respiration were taken while the fish were being chased, so presumably the fish were chased for at least 10 min each. This is very long for small 3-9 gram fish. What is the reasoning behind these long chase times?

The fish were chased until no attempts to escape were made as per standard protocol (e.g. Raby et al 2020); this timeframe was, on average, 10 minutes. This duration was not pre-selected; it is what the fish decided. Goldfish is a relatively aerobically powerful (e.g. Round Goby is not a very good swimmer), and these fish were somewhat large (~ 5.76 g average) compared to common models (e.g. zebrafish are ~ 1 g); this may be why the chase time was long time..

Page 7, lines 4-10: It says here that "Post-chasing, fish were immediately moved to respirometers with minimal air exposure, which has been shown to improve the accuracy of MMR measurements using the chase protocol (Raby et al., 2020). The respirometer was then immediately sealed and oxygen consumption measurements were taken in normoxia." If recordings of background respiration was ongoing while the fish were being chased, the respirometry chambers must have been sealed and needed to also be opened before the fish could be transferred there. How long did it take to transfer the fish from cessation of chasing? And where was the fish while chambers were being opened?

The fish were transferred as quickly as possible (approx. a maximum of 15 seconds to open chambers, net and place fish in chamber, and seal chamber). The fish were kept in the arena while the valves were being opened (~ 5 s).

Page 7, line 13: This section is about measuring background respiration, not correcting for it. That comes later.

The subtitle of the section has been updated accordingly.

Page 7, lines 28-30: How were fish exposed to 26 and 30°C ? Directly (acutely) from 22°C , or gradually warmed at a specified rate? This is important information that must be included.

The fish were acutely exposed to 26 and 30°C.

Page 8, Eq. 1 and Eq. 2: There is no definition of 'm' (it is obviously body mass but ought to be defined).

We've corrected this.

Page 9, Eq. 3: See general comment above about log-transformation. Also, the constants beta 1 and beta 2 are not defined (whereas beta 0 is).

Betas are not constants, they represent the estimated mean effect size coefficients for each respective parameter estimated by the model (as they would be notated in frequentist models as well); we have clarified this in the methods.

Page 9, line 17: It is unclear how the mass-adjustment becomes 'per kg' given that MO₂ values (cf. Eqs. 1 and 2) are divided by body mass, and body mass is in grams, not kg, according to the supplementary data table.

Thank you for pointing out this error. We incorrectly modeled our data using g, not kg, due to having our values in g in our dataset as you pointed out. However, our data Figure 1 used the proper conversion as the 'mass adjusted' columns converted 1 g to 0.001 kg during calculation. We have fixed this in our updated model, where, following your suggestion, we log-transformed MO₂ and mass. To avoid confusion in our data file, we have replaced the fish.weight column in the data file to one that is called 'mass' (in g).

Page 9, lines 17-21: The authors present differences in metabolic rates between the two experimental temperatures here. It would be more informative to present values for each temperature separately.

This is a great suggestion and we have modified Figure 1 to be a comparison between SMR and MMR at each temperature.

It is also unclear what is meant by "differences in the average of the mass-adjusted aerobic scope at 26°C and 30°C was negligible (179.5 mgO₂kg⁻¹h⁻¹ and 217.4 mgO₂kg⁻¹h⁻¹ for 26°C and 30°C respectively)." There is only one aerobic scope measurement at each temperature, so what are these 'differences in the average of the mass-adjusted aerobic scope'?

We apologize for the confusion, the line has been rephrased to be clearer.

Page 9, lines 54-55: Aerobic scope is not a "widely accepted proxy of fitness". See Clark et al. (2013) and Jutfelt et al. (2018).

The line has been removed.

Page 10, line 8: Raby et al. (2016) is not in the reference list.

We apologize for the typo, it should be 2020 not 2016.

Page 10, lines 8-10: The "ecological relevance of aerobic scope" is also not set in stone. Please consider the debate about aerobic scope and OCLTT (again, see Jutfelt et al. 2018 for summary).

We thank you for the excellent resource - however, we decided to take a different route and instead focus on the links between invasion biology and metabolism in our introduction/discussion. For instance, there is empirical evidence to support AS is predictive of feeding rate (one of the most, if not THE most, important metric predicting invasion impacts in the wild with empirical validation). The OCLTT debate is so large that we feel it will derail our piece. Generally speaking though we agree with Jutfelt et al. that the OCLTT is not well-supported empirically in all contexts.

Page 11, lines 19-23: The authors state here that "if Goldfish are acclimated to warmer conditions (i.e. throughout summer), it might be possible that the aerobic scope would further increase at the higher test temperature." Is it not more likely that aerobic scope would decrease with longer acclimation time, as is generally seen when ectotherms thermally compensate / acclimate (e.g. Norin et al. 2014)?

We have removed this as it was somewhat unclear. Although Ferreira et al. (2014) seemed to suggest that, at high acclimation temperatures, thermal optima seem to peak around acclimation temperatures (i.e. beneficial acclimation, sensu Angilletta). You've raised an interesting point though (acclimation duration -> peak AS?), one that we will consider going forward in our future experiments.

Page 22, Fig. 1. It would be good to calculate and add Q10s for the different metabolic rates. It would perhaps also make sense to plot RMR and MMR in the same panel, which would make it easier to visualize aerobic scope.

We thank the reviewer for the suggestion. However, we feel Q10 is out of the scope of our main question (and we've considerably shortened the manuscript). We also feel that having temperature treatments on the x-axis for comparison reasons is clearer because our main question and model analysis focused on comparing 26 and 30°C.

Cited literature (that is not already in reference list):

Fry FEJ & Hart JS (1948) The relation of temperature to oxygen consumption in the goldfish. The Biological Bulletin 94, 66-77. We have cited this.

Jutfelt F, Norin T, Ern R, Overgaard J, Wang T, McKenzie DJ, Lefevre S, Nilsson GE, Metcalfe NB, Hickey AJR, Brijs J, Speers-Roesch B, Roche DG, Gamperl AK, Raby DG, Morgan R, Esbaugh AJ, Gräns A, Axelsson M, Ekström A, Sandblom E, Binning SA, Hicks JW, Seebacher F, Jørgensen C, Killen SS, Schulte PM & Clark TD (2018) Oxygen- and capacity-limited thermal tolerance: blurring ecology and physiology. Journal of Experimental Biology 221, jeb169615.

We have decided to avoid discussing the OCLTT in great detail because we feel it derails from our focus, which is invasive species biology. Instead, we reframed our focus to discuss the plausible links between metabolism and invasion biology.

Killen SS, Christensen EAF, Cortese D, Závorka L, Norin T, Cotgrove L, Crespel A, Munson A, Nati JJH, Papatheodoulou M & McKenzie DJ (2021) Guidelines for reporting methods to estimate metabolic rates by aquatic intermittent-flow respirometry. Journal of Experimental Biology 224, jeb242522.

We have included the checklist in our supplemental and have cited this appropriately in methods.

Steffensen JF (1989) Some errors in respirometry of aquatic breathers: how to avoid and correct for them. Fish Physiology and Biochemistry 6, 49-59.

We have cited this.

Reviewer 2:

This study addresses a timely and ecologically relevant question regarding the physiological capacity of invasive goldfish to tolerate high water temperatures, with potential implications for their spread under climate change. The authors use aerobic scope as a physiological proxy to infer potential fitness at two elevated temperatures, building a narrative around the ecological risk posed by thermally tolerant invaders. However, there are several conceptual and methodological issues that weaken the study and conclusions, particularly around the interpretation of aerobic scope, experimental design, and data analysis. These should be addressed for the manuscript before being publishable.

We thank the reviewer for these thorough and thoughtful comments. We agree with your comments and believe that making the suggested edits has substantially improved our paper. We've responded to your comments in red, noting where we have made changes in the manuscript.

The manuscript refers to aerobic scope as a proxy for energy able to be allocated for reproduction, growth, and other fitness-related traits. However, this claim is often stated more than it is actually observed, and references would help support the argument that there is a link between aerobic scope and fitness. Also, aerobic scope represents the difference between maximum and standard (or occasionally routine) metabolic rate and reflects capacity, not actual energy allocation or utilization.

We thank you for the clarification on the definition of aerobic scope. We have revised the manuscript and have referenced studies to establish the link between aerobic scope and invasion success, without overstating the ecological relevance of AS. Specifically, we've reframed this to be more relevant to invasion ecology (where a small but growing body of work aims to link aerobic metabolism with invasion characteristics) and have approached our conclusions more conservatively. We hope our admittedly small study now reads as a first-look, empirical study of the aerobic metabolism of this feral population of Goldfish under different thermal conditions.

The use of RMR instead of SMR to estimate aerobic scope is problematic. RMR includes spontaneous activity and can differ substantially from SMR, especially if activity levels change across temperatures. If fish were less active at 30°C, for instance (as may occur if they are truly aerobically compromised), the RMR might decrease despite elevated metabolic demands, artificially inflating aerobic scope estimates. The manuscript should either justify the use of RMR over SMR or reframe the interpretation of the data accordingly.

We initially chose to consider our measurements "RMR" rather than "SMR" in order to be transparent about our data, mainly due to the fact that we did not measure MO₂ during a strict "rest" period for our fish (we have observed that our study goldfish are inactive during lights off in the evening, with minimal movement or signs of awareness if we quietly move around the lab; goldfish are also known to be diurnal). Basically, we fell into the group that Chabot et al. (2016) mentioned: "...many authors prefer the term routine MR (RMR), which includes a minor cost of activity..."

Because we measured fish in the morning (from 10-11 am) due to facility regulations prohibiting nighttime experimentation, we assumed they were not "at rest" using the strictest definition of "rest".

Ultimately, we did not want to mislead readers into assuming that we measured true SMR in the strictest sense of the term, although, to our knowledge, most fish metabolism studies do not actually ensure total inactivity/rest when SMR is measured. At the same time, we did use the quantile method to estimate RMR, which can help minimize MO₂ variation caused by 'minor costs of activity'. In light of your comment, we have made explicit the definition of RMR in our methods with reference to the inconsistent use of the terms RMR/SMR.

The experimental design introduces a serious temporal confound: all fish were measured at 26°C first, followed by 30°C a week later. It is therefore unclear whether the observed differences between temperatures are due to warming or extended lab acclimation. A counterbalanced or randomized temperature order would have been preferable. As is, any interpretation of temperature effects must be cautious and this caveat should be emphasized in both methods and discussion.

We agree this is an experimental limitation, as you and Reviewer 1 have both pointed out. We discuss the possibility that fish who were previously tested at 26°C later acclimatized to 30°C in our discussion as an experimental limitation. At the same time, exposure to a previous warm temperature before a second, warmer temperature may represent an ecologically realistic scenario.

The use of a short acclimation period (stated as "at least 1 week") prior to the 30°C trial is ambiguous. It's unclear whether all fish were given the same duration of acclimation, and

whether this is sufficient to assess steady-state physiological responses.

We have clarified that the interval between 26/30°C measurements was exactly 10 days for all fish (we previously reported this in our data sheet). We agree that the possibility of short-term acclimation between tests exists and discuss this as a limitation (as above). However, we note that the repeatability of RMR measurements is inversely related to the time period between measurements (White et al., 2013; <https://journals.biologists.com/jeb/article/216/10/1763/11540/The-repeatability-of-metabolic-rate-declines-with>; Norin & Malte, 2011; <https://journals.biologists.com/jeb/article/214/10/1668/10239/Repeatability-of-standard-metabolic-rate-active>) Therefore, we avoided longer interval times to balance (1) reducing the potential for short-term acclimation to warm exposure to 26°C and (2) obtaining a fair comparison of RMR after a 10-day interval. We note that cellular heat-shock responses typically resolve rapidly after acute heat exposures in fish (e.g., Podrabsky and Somero [2004] show that heat-shock transcripts are downregulated within 5 days of exposure to cycling temperatures in Killifish; a second study by Lewis et al. in salmonids showed transcripts disappeared within 8 h; a review by Massey & Hutchings [2020] showed whole-organism responses overwhelmingly do not persist in ectotherms, even with repeated heat shocks.) We have cited these studies to explain our choice in the methods.

Measurement of SMR/RMR lacks detail. The authors cite a 1.5-hour acclimation period in the respirometer, yet most literature suggests that SMR (or even stable RMR) is only achieved after much longer habituation periods (typically 5-24 hours). The relatively short chamber time may result in elevated MO₂ readings due to stress or activity, underestimating AS. This limitation should be acknowledged clearly.

At the onset of the experiment, we did a small pilot trial to inform us of the period after which RMR stabilizes at a low (minimum) value, which we mentioned in RMR section of the methods. Briefly, over the course of 3 h using 4 fish, we allowed DO to reach 80% in any one chamber. We then flushed all chambers manually until DO reached 100%, and repeated until 3h was completed. Using a rolling regression for 5-minute intervals during each measurement phase, we determined that the lowest MO₂ values occurred 72 minutes, on average, after initially placing fish in the respirometer. We understand that our pilot was limited by the short timeframe and small sample size, but we selected 3 hours as a reasonable maximum duration to balance the experimental workload for our lead author and facility security (we were not able to regularly use the animal facility in the evenings). Our animal care committee requires that experimenters are present at all times during experiments, so longer timeframes were not feasible for us.

The paper should clarify whether closed or intermittent-flow respirometry was used. While the text implies closed respirometry, the protocols and calculations would benefit from further clarification.

We have specified that our measurements were done using closed respirometry. As Reviewer 1 also mentioned, we understand that our methods are somewhat “hybrid” because we flushed chambers gently during acclimation. Due to equipment limitations we did not have access to a timed pump setup for true intermittent flow; we felt that we could minimize buildup of nitrogenous wastes during acclimation by pumping with a simpler design. We have explained this more clearly in the methods.

The distinction between mass-specific and mass-adjusted metabolic rates is blurred. The manuscript describes values as “mass-adjusted,” yet it appears the rates are simply expressed per kg (by dividing RMR by mass), without statistical correction (mass-specific metabolic rate decreases with body mass). If linear models were used to adjust for mass, then the estimates reported should be model outputs or adjusted residuals.

We thank you for pointing out this error. To clarify, in Fig. 1, we presented data that were simply divided by mass (i.e., per-kg MO₂). However, in our model, we followed the recommendation of Chabot & Farrell (2016) to include mass as a covariate, which was appropriate as our fish were all in the same life stage (age) but had different masses. As a result, our Fig. 2 and model analyses are mass-corrected following standard practice. We note that we have also updated our model to

log-transform both MO₂ and mass to account for potential nonlinearity, as suggested by Reviewer 1. Based on your suggestion and the suggestion of Reviewer 1, we have presented data in Fig. 1 as mass-standardized using the methods you outlined (and following Norin et al. (2015)), where all individuals were mass-corrected to the average mass of 5.76 g by using model outputs to determine residuals for (raw) size data of individual fish.

The quantile method (q0.25) for RMR is fine in principle, but the authors do not report the number of slopes used or whether these were independent. If slopes are tightly clustered in time (which I think they must have been because they seem to be using closed respirometry), the quantile estimate may be artificially narrow due to temporal autocorrelation.

We have reported the number of slopes used in our methods section, and we agree that they are clustered in time. In an attempt to be more conservative with our RMR estimates, we chose to use q.25 rather than smaller quantiles (e.g. q.01). We believed this choice of a larger 25% quantile struck a balance between accounting for possible periods of minor activity, while also not over-correcting our MO₂ data.

The manuscript states that "little is known" about temperature effects on aerobic scope in goldfish. This is inaccurate a several prior studies have examined these relationships, including:

We apologize, we meant "wild" Goldfish (i.e. not pets/commercial aquaculture). We intended to focus on the invasive species context here and have clarified this in the text. Nevertheless, we agree that its a huge omission not to include these studies, and as such we have described and cited them where appropriate in the text. Thank you for providing these citations.

E.G. Ferreira, E. O., Anttila, K., & Farrell, A. P. (2014). Thermal optima and tolerance in the eurythermic goldfish (*Carassius auratus*): relationships between whole-animal aerobic capacity and maximum heart rate. *Physiological and Biochemical Zoology*, 87(5), 599-611.

Pang, X., Cao, Z. D., & Fu, S. J. (2011). The effects of temperature on metabolic interaction between digestion and locomotion in juveniles of three cyprinid fish (*Carassius auratus*, *Cyprinus carpio* and *Spinibarbus sinensis*). *Comparative Biochemistry and Physiology Part A: Molecular & Integrative Physiology*, 159(3), 253-260.

The introduction and discussion should be revised to more accurately reflect the existing literature.

We thank you for these references and have better-integrated our results in the context of the literature - our discussion is briefer and more realistic about the implications of our work. We do note that both previous studies used cultured fish, whereas we captured wild invasive fish locally; therefore, we believed we may see differences in aerobic metrics (possibly due to local plastic or genetic adaptation) and feel our results may be more relevant than previous studies to conservation and invasion biology. We have acknowledged existing studies and shifted the perspective of the manuscript to focus more on wild populations in our introduction and discussion.

Resubmission

First decision letter

MS ID#: bio.062160

MS TITLE: Invasive Goldfish (*Carassius auratus*) maintain aerobic scope across acute warm water temperatures

AUTHORS: Nazeefa Arifina Nashrah; Nicholas Edward Mandrak; Melanie Duc Bo Massey

I have now reached a decision on the above manuscript.

The reviewer reports are shown at the bottom of this email or can be accessed, together with a copy of this decision letter, by going to:

As you will see, the reviewers raised a number of substantial criticisms that prevent me from accepting the paper at this stage. In particular, they both raise very substantial concerns about the lack of transparency in the Methods. This submission is a borderline case for rejection, but after consideration I have decided to give the opportunity to revise and resubmit. This is no way guarantees publication - I shall be reviewing the manuscript once more upon resubmission and it is crucial that Methodological weaknesses, alongside other manuscript issues raised by the reviewers, are dealt with either through edits to the manuscript and/or rebuttal explanations.

Please ensure that you clearly highlight all changes made in the revised manuscript. Please avoid using 'Tracked changes' in Word files as these are lost in PDF conversion.

I should be grateful if you would also provide a point-by-point response detailing how you have dealt with the points raised by the reviewers in the 'Response to Reviewers' box. Please attend to all of the reviewers' comments. If you do not agree with any of their criticisms or suggestions please explain clearly why this is so.

Reviewer 1

Comments to Author

General comments

This paper aims at testing the effect of two warm temperatures (26 °C and 30 °C) on the aerobic scope of goldfish, within the context of global warming. While there are previous studies on the effect of high temperature on the metabolism of this species, the potential value of this paper is in the use of wild caught goldfish and the context of invasive populations in a warming climate. However, there are many weak parts, especially in the methods, that prevent me from accepting this paper as such. I recommend a clearer description of the methods as well as acknowledgement of the many limitations of this work (i.e. use of RMR instead of SMR, short acclimation for RMR measurements, etc.). This paper is a borderline case for which extensive revision is necessary.

Specific comments

Introduction

1) Applying aerobic scope to predict distributions. There a number of experimental, modeling and review papers that I strongly recommend to be taken into account when introducing this idea. These are Farrell et al 2008, Marras et al 2015; Teal et al 2018.

2) In addition, the OCLTT is presented here as the accepted theory, while much debate has occurred around such a theory. The authors should at least mention this debate in the introduction and in the discussion, as a potential limitation of this work.

3) The introduction should clarify the definition of AS, SMR and RMR. SMR is brought up only in page 10 and not defined. In addition, a strict measurement of AS is based on the difference between MMR and SMR (Svendsen et al 2016, Roche et al 2013). While previous work has in some cases used MMR-RMR to estimate AS, the authors should discuss the limitations of using RMR rather than SMR, and the potential confounding effect of activity on RMR at different temperatures.

Methods

4) Page 7: Maximum density of 33 individuals, please indicate the range.

5) Page 8: The chamber volume was about 50 times the volume of the fish. (Average mass of 5.76 g). However, fish mass varied considerably (given 4-7 cm range in length), therefore the ratio of the chamber volume /fish volume also must have varied a lot. Svendsen et al (2016) recommend

20-50 as the ratio suggested. Here, the range should be indicated, and it will certainly exceed 50 in its upper limits. This limitation should be acknowledged in the discussion.

6) Page 9: More details are needed about the acute exposure. Were fish simply dropped in water with higher temperature, directly, with a delta T occurring within seconds?

7) Page 9: After testing at 26°C, fish were allowed to rest for 10 days. Please indicate at what temperature. (22°C?)

8) Page 9: Intermittent flow respirometry is the preferred method used (Svendsen et al 2016). Therefore the limitation of the methods used here (i.e. potential disturbance of the fish) in this regard should be acknowledged

9) Page 9: More information about the standardization is needed. Simply dividing by body mass does not work because AS does not scale linearly with size. More information is needed about how to deal this potential issue.

10) Page 10: RMR includes minor costs of activity. However, such an activity can depend on temperature levels. The authors should either show results on such an activity (i.e. based on video recordings) or discuss this limitation.

11) Page 10. An acclimation period of 90 minutes to obtain RMR is quite short given all the disturbance the fish experienced. While there are species-specific difference, typically a much longer period is used, up to 24 hours. The authors should show data from the pilot trials mentioned, in which respiration rates reached a plateau. These trials only lasted 3 hours so it is not clear if metabolic rates may have declined further, after these 3 hours.

Discussion

12) Page 17: Underestimation of AS, as a result of chase methods are mentioned here. In addition to this limitation, AS may be underestimated when using RMR rather than SMR. This and the potential effect of temperature on activity should be discussed in depth.

13) Page 17: The limitation of the use of 26°C exposure before the 30°C needs to be discussed more in depth. The authors claim ecological realism here. However, it is not clear if the fish was exposed to the sequence 22, 26, 22, 30°C or any other sequence, and its relative realism or lack thereof, needs to be discussed.

Figures.

14) Fig 1. Perhaps I have missed something here, possibly due the standardization process. Subtracting the values of RMR from MMR, I obtain values that are much lower than the AS shown in the figure. Please clarify.

Reviewer 2

Comments to Author

I appreciate the authors' efforts to revise the manuscript and respond to reviewer comments. The revised manuscript is improved in several respects, and I commend the authors for engaging with prior critiques in a constructive manner. However, I remain concerned that the clarity and transparency of the methods section are still insufficient, particularly in ways that hinder reproducibility and confidence in the results. In my view, this issue now represents the main barrier to publication. My detailed comments are below.

1) My main concern is that the methods section remains difficult to follow, even on a second reading. As it stands, critical information about the experimental setup, timing, and procedures is scattered, sometimes only appearing later in the discussion or figure captions. This creates confusion and undermines transparency. The full timeline of fish holding, testing at different

temperatures, and acclimation intervals is unclear and can only be pieced together from scattered mentions.

2) Background respiration in particular is only briefly addressed and is very unclear. It remains unclear how it was measured, how consistently across trials or chambers, and how it was incorporated into slope corrections. Was there a parallel empty chamber? What does it mean that background was measured while the fish were being chased? So this means the blank was only around 10 minutes? How was this subtracted in parallel for the entire duration of the trial given that the blank will increase over time?

3) The definition of RMR in the intro is still unclear or incorrect, as it is described as being the costs of maintenance. I understand why you're not calling it SMR, but if that's the case then don't define it as such.

4) While toned down compared to the original version, some claims about the ecological relevance of the observed metabolic changes still overreach given the limited temperature range and constrained aerobic scope estimate. I suggest adding further caveats in the discussion.

Reviewer's Responses to Questions

Experimental quality

Does each figure have the proper controls?

If 'No', please indicate reasons in Comments for Author box below.

Reviewer #1:

- Yes

Reviewer #2:

- Yes

Were the data analyzed using appropriate statistical tests?

If 'No', please indicate reasons in Comments for Author box below.

Reviewer #1:

- Yes

Reviewer #2:

- Yes

Reproducibility

Were experiments performed using adequate number of biological replicates?

If 'No', please indicate reasons in Comments for Author box below.

Reviewer #1:

- Yes

Reviewer #2:

- Yes

Does the methods section provide sufficient detail to permit reproducibility?

If 'No', please indicate reasons in Comments for Author box below.

Reviewer #1:

- No

Reviewer #2:

- No

Completeness

Are the manuscript's conclusions supported by the data?

If 'No', please indicate reasons in Comments for Author box below.

Reviewer #1:

- Yes

Reviewer #2:

- Yes

Scholarship

Do the authors cite and discuss the merits of data that would argue for and against their conclusion?

If 'No', please indicate reasons in Comments for Author box below.

Reviewer #1:

- No

Reviewer #2:

- Yes

Does the manuscript title & abstract accurately reflect the contents of the manuscript, without hyperbole?

If 'No', please indicate reasons in Comments for Author box below.

Reviewer #1:

- Yes

Reviewer #2:

- Yes

First revision

Author response to reviewers' comments

Reviewer 1:

General comments

This paper aims at testing the effect of two warm temperatures (26 °C and 30 °C) on the aerobic scope of goldfish, within the context of global warming. While there are previous studies on the effect of high temperature on the metabolism of this species, the potential value of this paper is in the use of wild caught goldfish and the context of invasive populations in a warming climate.

However, there are many weak parts, especially in the methods, that prevent me from accepting this paper as such. I recommend a clearer description of the methods as well as acknowledgement of the many limitations of this work (i.e. use of RMR instead of SMR, short acclimation for RMR measurements, etc.). This paper is a borderline case for which extensive revision is necessary.

We thank the Reviewer and are grateful for their comprehensive constructive feedback. The suggestions you have given us are very helpful for improving our paper and our methods for future studies. We have responded to your comments in red, noting where we have made changes in the manuscript.

Specific comments

Introduction

1) Applying aerobic scope to predict distributions. There are a number of experimental,

modeling and review papers that I strongly recommend to be taken into account when introducing this idea. These are Farrell et al 2008, Marras et al 2015; Teal et al 2018.

We thank you for these references and have integrated them into our introduction (lines 94-95), where we discuss the potential links between AS and invasion biology, to strengthen our introduction of using AS to predict distributions.

2) In addition, the OCLTT is presented here as the accepted theory, while much debate has occurred around such a theory. The authors should at least mention this debate in the introduction and in the discussion, as a potential limitation of this work.

Although we previously decided to focus on the links between invasion biology and metabolism in our introduction/discussion rather than discussing the OCLTT debate since it is so vast, we have revised the introduction to mention differing views on OCLTT and then brought the focus back to the potential relationship between aerobic scope and invasion physiology (80-93).

3) The introduction should clarify the definition of AS, SMR and RMR. SMR is brought up only in page 10 and not defined. In addition, a strict measurement of AS is based on the difference between MMR and SMR (Svendsen et al 2016, Roche et al 2013). While previous work has in some cases used MMR-RMR to estimate AS, the authors should discuss the limitations of using RMR rather than SMR, and the potential confounding effect of activity on RMR at different temperatures.

We have revised the definitions of all three terms including use of SMR in the introduction (Lines 70 - 79). We have also discussed the limitations of using RMR in the discussion.

Methods

4) Page 7: Maximum density of 33 individuals, please indicate the range.

We have included the range in line 162; we corrected an error in reporting holding density during quarantine such that we now specify the density was 22 - 25. The reason for the error was that a different cohort of fish was being quarantined for another student's experiment (the maximum density of all quarantine tanks was 33, but we only took fish from tanks with densities of 22 - 25).

5) Page 8: The chamber volume was about 50 times the volume of the fish. (Average mass of 5.76 g). However, fish mass varied considerably (given 4-7 cm range in length), therefore the ratio of the chamber volume /fish volume also must have varied a lot. Svendsen et al (2016) recommend 20-50 as the ratio suggested. Here, the range should be indicated, and it will certainly exceed 50 in its upper limits. This limitation should be acknowledged in the discussion.

We have reported the range of fish:respirometer volumes in the methods (31 - 112 times), but have also acknowledged that authors report different ideal ranges (e.g., in Clark et al. (2013), Aerobic scope measurements of fishes in an era of climate change: respirometry, relevance and recommendations, the authors suggest between 1:20 and 1:100). Additionally, Svendsen et al. (2016) recommends a hard limit of a chamber no more than 200 times the volume of the fish. We have added additional citations to these studies. Our range appears to be within typical limits in the field.

6) Page 9: More details are needed about the acute exposure. Were fish simply dropped in water with higher temperature, directly, with a delta T occurring within seconds?

Yes, the fish were netted from their respective tanks (at 22C) and placed inside the respirometer chambers already set to test temperatures. Although a gradual increase in temperature would be more ideal to reduce chances of thermal shock, we were restricted by the inability to use the facility in the evenings, preventing us from lengthening the duration of our trials. We discuss this later and emphasize that the results of our experiments are reflective of a "heat shock" rather than gradual warming.

7) Page 9: After testing at 26 °C, fish were allowed to rest for 10 days. Please indicate at what temperature. (22 °C?)

We have altered the methods and in the supplemental flow chart to make this clearer. Fish were allowed to rest for 10 days at a temperature of 22 °C (under the previous common-garden acclimation conditions).

8) Page 9: Intermittent flow respirometry is the preferred method used (Svendsen et al 2016). Therefore the limitation of the methods used here (i.e. potential disturbance of the fish) in this regard should be acknowledged.

We have acknowledged this as a limitation in our discussion, but we would like to say we were unable to use automated intermittent flow due to financial constraints (costs for premade setups can be upwards of \$20,000 CAD). Static respirometry is still a widely used and valid method. With respect to using closed respirometry, we followed best practices (e.g. including a stir bar for water mixing, selecting an appropriate chamber size for the range of fish we had) and even sought to improve upon previous methods (flushing chambers with a custom inexpensive pump design, using a pilot experiment to determine a fair acclimation period). Some of the issues (e.g. disturbance) may not even be fully eliminated using an intermittent-flow design, given that pumps coming on and water pushing through the chamber can still disturb a fish. We have mentioned in our methods that we took great care when manually closing valves to not disturb fish.

However, we note that a previous reviewer suggested modifications to economically create an intermittent-flow type design, which we are looking into going forward.

9) Page 9: More information about the standardization is needed. Simply dividing by body mass does not work because AS does not scale linearly with size. More information is needed about how to deal with this potential issue.

The standardization was done to present data results in a way that corrects for variation in size of our experimental fish. In lines 314-320, we state that we determined residuals for each fish's AS, RMR, and MMR values based on the coefficients of our model using mass as a covariate, and applied these residuals to standardize AS, RMR, and MMR values "back to that" of an average 5.76 g fish (as in the detailed explanation in our reference, Norin et al. (2015); <https://doi.org/10.1111/1365-2435.12503>, see the paragraph preceding "Results"). We summarized this process in our paper in brief but included the reference as we felt Norin et al. explain the standardization methods very clearly. This standardization was recommended by both reviewers of our previous draft as a means of dealing with nonlinearity of the mass-metabolism relationship.

We note, however, that the standardization is for visual purposes (to display our data in, effectively, a mass-independent way). The model results (Fig. 2) from our model in which mass was used as a covariate have a nonlinear mass-correction applied.

10) Page 10: RMR includes minor costs of activity. However, such an activity can depend on temperature levels. The authors should either show results on such an activity (i.e. based on video recordings) or discuss this limitation.

We have acknowledged this as a limitation in our discussion.

11) Page 10. An acclimation period of 90 minutes to obtain RMR is quite short given all the disturbance the fish experienced. While there are species-specific differences, typically a much longer period is used, up to 24 hours. The authors should show data from the pilot trials mentioned, in which respiration rates reached a plateau. These trials only lasted 3 hours so it is not clear if metabolic rates may have declined further, after these 3 hours.

We agree that testing a longer acclimation period would have been preferable, but our facilities regulations do not permit researchers to access the room at night for security reasons; further, when animals are in experimental situations (i.e. within the respirometer), our Animal Care

Committee requires that researchers are present at all times to supervise animals (making a long acclimation period unfeasible in this, or any other experiment at our facility, going forward). Given these limitations, we conducted the pilot experiment to find the best acclimation time that balanced logistics. We have added an illustrative example of how the analysis for the pilot was done in our supplemental material to help clarify this.

Given that we used a 3 h acclimation test period in trials and found that, over the 3 h, the lowest, stable ($R^2 > 0.95$) rates of oxygen consumption occurred after 72 min on average (and not in the vicinity of 180 min), we feel that our pilot trial was valid (at least given the logistical constraints preventing us from using much longer acclimation periods).

Discussion

12) Page 17: Underestimation of AS, as a result of chase methods are mentioned here. In addition to this limitation, AS may be underestimated when using RMR rather than SMR. This and the potential effect of temperature on activity should be discussed in depth.

We thank the Reviewer for the suggestion and have discussed the limitation.

13) Page 17: The limitation of the use of 26°C exposure before the 30°C needs to be discussed more in depth. The authors claim ecological realism here. However, it is not clear if the fish was exposed to the sequence 22, 26, 22, 30°C or any other sequence, and its relative realism or lack thereof, needs to be discussed.

We thank the Reviewer for the suggestion. You have interpreted correctly and, with your suggestion, we have discussed this limitation, stating that our results should be interpreted in light of our methods - that they reflect responses to acute heat shocks.

Figures.

14) Fig 1. Perhaps I have missed something here, possibly due the standardization process. Subtracting the values of RMR from MMR, I obtain values that are much lower than the AS shown in the figure. Please clarify.

You are correct - the difference occurred because of the standardization, which was requested by previous reviewers, to correct the visual display of data for Fig. 1.

The reason is that our models indicated the nonlinear relationship between mass-aerobic scope differs from the nonlinear relationship between mass-RMR and mass-MMR (this was mentioned in Results lines 340+). Based on the methods specified by Norin et al. (2015), we applied standardization to each of RMR, MMR, and AS models independently (each model used whole-organism metabolic rates corrected with mass as a covariate). As a result, when we standardized the masses, there was a different mass scaling coefficient used for each of RMR, MMR, and AS, which means in our mass-standardized figure, $MMR - RMR$ does not necessarily equal AS.

Actually, the different nonlinear mass scaling of RMR/MMR/AS revealed by the models - i.e., the scaling coefficient of $AS > MMR > RMR$ -- is itself an interesting finding worth pursuing in a more detailed manner in the future (but not the goal of this piece and we are uncertain of the ecological implications of this finding).

Reviewer 2:

I appreciate the authors' efforts to revise the manuscript and respond to reviewer comments. The revised manuscript is improved in several respects, and I commend the authors for engaging

with prior critiques in a constructive manner. However, I remain concerned that the clarity and transparency of the methods section are still insufficient, particularly in ways that hinder reproducibility and confidence in the results. In my view, this issue now represents the main barrier to publication. My detailed comments are below.

We thank the Reviewer for their kind and constructive feedback and for agreeing to re-review our piece. We have incorporated their suggestions to ensure transparency and clarity of the methods, and have responded to their comment in red. In addition to clarifying details of the methods, we have added a flow chart to the methods to make it easier for readers to understand the methods more clearly.

1) My main concern is that the methods section remains difficult to follow, even on a second reading. As it stands, critical information about the experimental setup, timing, and procedures is scattered, sometimes only appearing later in the discussion or figure captions. This creates confusion and undermines transparency. The full timeline of fish holding, testing at different temperatures, and acclimation intervals is unclear and can only be pieced together from scattered mentions.

We thank the Reviewer for the suggestion and in response, to make it easier for our readers to understand, we have added a flow chart to the supplementary material to organize the information and have also rewritten the methods section to make it easier to understand in a chronological order.

2) Background respiration in particular is only briefly addressed and is very unclear. It remains unclear how it was measured, how consistently across trials or chambers, and how it was incorporated into slope corrections. Was there a parallel empty chamber? What does it mean that background was measured while the fish were being chased? So this means the blank was only around 10 minutes? How was this subtracted in parallel for the entire duration of the trial given that the blank will increase over time?

We apologize for the confusion. To clarify, background respiration was measured in each individual respirometer, with "fish water" from individual fish, for each trial, in between RMR and MMR trials.

We chose not to use a parallel blank because it may not accurately reflect microbial buildup resulting from the fish itself.

As each fish underwent the chase protocol, their respective chamber, filled with water that the fish was in, was sealed and oxygen consumption was measured for approximately 10-15 minutes (the time including the chase protocol set-up period and period during which the fish was chased varied between 10-15 minutes on average for each individual).

After the chase protocol was complete for an individual fish, the fish was returned to its original chamber to record MMR. We felt this would result in the fairest measurement of the actual microbial conditions in each fish's chamber (rather than averaging a blank across all chambers), although in the future we could potentially increase this recording time by allowing fish to rest longer in a different chamber before exhaustive chase trials.

In data handling, we use the entire slope of the background respiration values for each individual chamber and subtracted that from both the RMR slope and MMR slope values from that same chamber (i.e. from an individual fish's RMR/MMR from that trial). In response to the Reviewer's comments, we have further clarified our methods and supplementary table S1 (the respirometry checklist from Killen et al., 2021) to be clearer.

3) The definition of RMR in the intro is still unclear or incorrect, as it is described as being the costs of maintenance. I understand why you're not calling it SMR, but if that's the case then don't define it as such.

We have revised the definition of the RMR in the introduction and revisited it in the discussion to ensure the reader is not misled; (i.e.: metabolic rate in the absence of digestion and growth, but not during quiescence, and may include very minor activity such as turning).

4) While toned down compared to the original version, some claims about the ecological relevance of the observed metabolic changes still overreach given the limited temperature range and constrained aerobic scope estimate. I suggest adding further caveats in the discussion.

We thank the Reviewer for the suggestion and have added further limitations in our manuscript.

Second decision letter

MS ID#: bio.062160R1

MS TITLE: Invasive Goldfish (*Carassius auratus*) maintain aerobic scope across acute warm water temperatures

AUTHORS: Nazeefa Arifina Nashrah; Nicholas Edward Mandrak; Melanie Duc Bo Massey

I have had the opportunity to read through your rebuttal and the edits you have made to your manuscript, and I am happy to tell you that your manuscript has been accepted for publication in Biology Open, pending our standard publication integrity checks. It was accepted on 29 July 2025.